# Promising Therapeutic Strategies for Colorectal Cancer Treatment Based on Nanomaterials

**DOI:** 10.3390/pharmaceutics14061213

**Published:** 2022-06-07

**Authors:** Natalia Krasteva, Milena Georgieva

**Affiliations:** 1Institute of Biophysics and Biomedical Engineering, Bulgarian Academy of Sciences, “Acad. Georgi Bonchev” Str., bl. 21, 1113 Sofia, Bulgaria; 2Institute of Molecular Biology “Acad. R. Tsanev”, Bulgarian Academy of Sciences, “Acad. Georgi Bonchev” Str., bl. 21, 1113 Sofia, Bulgaria

**Keywords:** colorectal cancer (CRC), nanomaterials, nanocarriers, nanoparticles, cancer therapy, targeted therapy, graphene, graphene oxide

## Abstract

Colorectal cancer (CRC) is a global health problem responsible for 10% of all cancer incidences and 9.4% of all cancer deaths worldwide. The number of new cases increases per annum, whereas the lack of effective therapies highlights the need for novel therapeutic approaches. Conventional treatment methods, such as surgery, chemotherapy and radiotherapy, are widely applied in oncology practice. Their therapeutic success is little, and therefore, the search for novel technologies is ongoing. Many efforts have focused recently on the development of safe and efficient cancer nanomedicines. Nanoparticles are among them. They are uniquewith their properties on a nanoscale and hold the potential to exploit intrinsic metabolic differences between cancer and healthy cells. This feature allows them to induce high levels of toxicity in cancer cells with little damage to the surrounding healthy tissues. Graphene oxide is a promising 2D material found to play an important role in cancer treatments through several strategies: direct killing and chemosensitization, drug and gene delivery, and phototherapy. Several new treatment approaches based on nanoparticles, particularly graphene oxide, are currently under research in clinical trials, and some have already been approved. Here, we provide an update on the recent advances in nanomaterials-based CRC-targeted therapy, with special attention to graphene oxide nanomaterials. We summarise the epidemiology, carcinogenesis, stages of the CRCs, and current nanomaterials-based therapeutic approaches for its treatment.

## 1. Introduction

Colorectal cancer (CRC) is a severe health problem with an enormous economic burden on society and the families of cancer patients [1,2,3]. With 1.93 million new cases per year and with 916,000 deaths, CRC is the third-most frequently occurring cancer globally [4]. Numerous treatment strategies to fight CRC are in use [3]. However, there are still no available approaches that completely eradicate this life-threatening disease. The reason for this is the fact that cancer therapy is currently mostly limited to classical methods like surgery, radiation, and chemotherapy. All techniques, though, have limitations and do not guarantee satisfactory results [5]. Among those confines is the poor solubility of the drug in an aqueous solution, drug resistance, non-specific targeting, low-retention effectiveness, and adverse effects on normal cells. Therefore, novel therapeutic strategies are constantly needed to improve the safety and effectiveness of cancer therapy and overcome its drawbacks [6]. 

In recent years, new methods based on nanomaterials have shown great promise in anticancer treatment [7]. Many nanomaterials have been proposed and are under investigation in cancer therapy, and they include polymeric nanoparticles, micelles, liposomes, mesoporous inorganic nanoparticles, metal oxides, noble materials, and carbon nanomaterials [8]. Most of the nanoparticles were designed as carriers in drug delivery systems [9] or for use in external stimuli-based systems, such as photodynamic therapy (PDT), [10] photothermal (PTT) [11,12] and magnetic therapy [13]. Besides the above-mentioned applications of nanomaterials, the newly proposed two-dimensional nanomaterials have caught the attention of many scientists as ultrathin materials with a high degree of anisotropy and chemical functionality [14]. Graphene oxide is among these newly developed nanoparticles. It has attracted significant global research interest since 2008 when Liu et al. demonstrated for the first time its capabilities as a drug carrier for a water-insoluble, aromatic anticancer drug (SN38). In these studies, graphene oxide was modified with polyethylene glycol [15]. Since then, the research on graphene oxide-based materials has opened up new perspectives toward the design and development of improved therapeutic efficiency of cancer treatments. 

Graphene oxide-based nanomaterials have unique properties, including chemical and mechanical stability, two-dimensional structures, and biocompatibility [7]. Moreover, they have a large and easy to the functionalized surface by linking with epoxide hydroxyl, carboxyl, and hydroxyl groups [7]. These groups can be further used to change the surface characteristics of GO and provide attachment sites to various molecules, including protein, deoxyribonucleic (DNA), and ribonucleic acids (RNA) [7]. Consequently, graphene oxide-based nanomaterials appear suitable for various applications, including drug or gene delivery platforms, phototherapy, and bioimaging [16,17,18], thus demonstrating great capacities in cancer treatments. 

Every year, a large number of research papers regarding graphene oxide-based nanomaterials in cancer are published, but there is still no review summarising the new findings of the application of graphene oxide-based nanomaterials in colorectal cancer strategy treatments [7]. Therefore, in this review, we summarise the most recent investigations of graphene oxide-based nanomaterials in anticancer therapy and, more especially, in colorectal cancer treatment. The epidemiology, carcinogenesis, stages of the CRCs, and current therapeutic approaches for colon cancer treatment are summarised, followed by the latest achievements of the most widely used nanomaterials in colorectal cancer treatment. We next review the synthesis, structure, and properties of graphene oxide-based nanomaterials and, finally, discuss their applications in anticancer strategies, specifically for CRC treatment.

## 2. Cancer—The Hidden Pandemics

### 2.1. Overview of the Current Situation Worldwide

Nowadays, cancer remains one of the main causes of death worldwide. It is one of the deadliest diseases in human history, with an extremely poor prognosis [4]. It is often defined as the “Pathology of the century” because every one in six deaths throughout the world is due to cancer [4,19]. Cancer is the generic term for a large group of diseases that can affect any part of the body [4]. There are more than two hundred distinct types of cancer, as it results from the abnormal proliferation of different types of cells in the body [20]. Cancers of the lung, female breasts, and colorectum are the top three cancer types in terms of incidence and are ranked within the top five in terms of mortality. Together, those three cancer types are responsible for one-third of the cancer incidences and mortality burden worldwide [4]. They arise from the accumulation of somatic mutations and other genetic changes in cells, which affect cell division checkpoints and result in both abnormal cell growth and, eventually, tumorigenesis. Signalling pathways are usually altered when cancer occurs, and the inhibition of physiological apoptosis is also observed, thus contributing to cancer development and resistance to radiotherapy and chemotherapy. Inflammation and immune system disorders are also related to cancer [20,21].

In 2020, 19.3 million new cancer cases and almost 10.0 million deaths due to cancer were registered according to the World Health Organization (WHO), making cancer the second cause of death worldwide [22]. The WHO predicts that, in 2040, the number of cancer patients will increase to 29.4 million per year [22]. Cancer patients are the hidden cost of the COVID-19 pandemics because millions of them did not receive anticancer treatment or were undiagnosed during this period. Based on the data collected from 17 European countries, the European Cancer Organisation (ECO) reported that approximately one hundred million screening tests were not performed, and more than one million Europeans are now living with undiagnosed cancer due to COVID-19 [23]. Moreover, a substantial increase in the number of avoidable cancer deaths is expected as a result of treatment and diagnosis delays, as well as due to other complications in response to the current pandemics [24,25]. Several studies have reported on the impact of delays and reduction in diagnosis on cancer survival outcomes in different Western European countries. A reduction in the number of cancer incident diagnoses between 12% in Sweden and 42% in Denmark was reported [26,27]. Data from the United States revealed a significant decline in incident diagnoses for six common cancer types [28]. The decrease in the incident diagnoses ranged from 25% for pancreatic to 52% for breast cancer, respectively [28]. Correspondingly, an increase in the number of cancer deaths for the four most common cancer types has been registered in England, compared with the pre-pandemic period [24].

### 2.2. Colorectal Cancer—An Old Man’s Disease

#### 2.2.1. CRC Epidemiology

Colorectal cancer (CRC) is the most devastating disease identified among all malignancies [29,30]. 10% of global cancer incidence and 9.4% of all cancer deaths result from CRC, thus making it the second-most malignant and deadly cancer in the world [1,2]. It is also classified as the second-leading reason for cancer-associated deaths in women and the third-highest common disease in men worldwide [29,30]. Globally, incidences vary 10-fold, with the highest rates in Australia, New Zealand, Europe, and the USA, while the lowest rates of CRC incidence have been reported in Africa and South-Central Asia [22,30]. CRC is more typical for higher economic status countries and elderly people over 50. According to an analysis performed in 2021, the CRC rates appeared to climb in younger adults with a low income [31,32,33,34]. Among all colon cancer patients, approximately 5–20% have a familial or congenital mutation(s) in the gene(s) that increases the colon cancer risk at an earlier stage of life [35]. The remaining majority (>80%) of CRC arises from sporadic mutations and occurs in people without a genetic predisposition or family history [36]. Sporadic CRC tends to develop in the later stage of life and is significantly influenced by external nongenetic risk factors, such as lifestyle, low physical activity, high-caloric and high-fat diet, obesity, smoking, alcohol use, etc. [37,38,39].

#### 2.2.2. Colon or Colorectal Cancer: What Is the Difference?

Colorectal carcinoma starts in the colon or the rectum. These cancers can also be called colon or rectal cancers, depending on where they start. Therefore, the two are combined in the term “colorectal cancer” (CRC) or are used interchangeably [40]. Indeed, colorectal cancer is the common name for colon and rectal cancers. They are often grouped because of the many common features, such as some clinical symptoms (bleeding, pain, and changes in stool), diagnosis, risk factors, and genetics. However, there are certain differences between colon and rectal cancers, and they include the place in the organisms where they start, disease recurrence, sex differences, invasion of nearby tissues, and treatment options (Figure 1) [41]. Both the colon and rectum are part of the large intestine (or large bowel), which is part of the digestive system, also called the gastrointestinal (GI) system. The colon is about 5 feet (150 cm) long and is divided into five major segments [42]. The rectum is the last segment. It is five to six inches long and connects the colon to the anus. The colon and rectum play important roles in digestion by aiding in the absorption of water, minerals, nutrients, and storage of wastes. The function of the colon is to absorb water and salt from the remaining food after it goes through the small intestine. The waste matter that is left after going through the colon goes into the rectum, and it is stored there until it passes through the anus [43,44].

Both the colon and rectum have a very similar histological structure typical for the digestive tract. Histologically, they contain the following layers: mucosa, submucosa, muscularis, and serosa/adventitia. The mucosa is the innermost layer (closest to the lumen), which lines the tract. It consists of epithelium, lamina propria, and muscularis mucosae. The intestinal epithelium contains goblet cells and columnar enterocytes. The lamina propria supports the epithelium and presents a loose connective tissue containing blood and lymph vessels. The muscularis mucosae is a muscle layer called lamina muscularis and is situated under the lamina propria. Thesubmucosa, on the other hand, is a connective tissue layer supporting the mucosa, while the muscularis externa is the muscular wall of the tract, surrounding the submucosa. Interestingly, the adventitia/serosa is the outermost, protective layer of the colon. At the lower end of the tract - at the anal zone, the epithelium becomes stratified into squamous non-keratinised epithelium [45,46]. Interestingly, the blood supply of the colon and rectum and their lymphatic drainage and innervation are distinctive. 

In the rectum, the serosa layer is absent which facilitates the tumour in breaking through and spreading locally. Additionally, the rectum is much shorter than the colon, making it easier for cancer to propagate to the surrounding tissues. Rectal cancer metastasis is potentiated by the blood supply, to which rectal cancer cells have access. Depending on the size and position of the tumour, rectal cancer can pose a threat to essential bodily functions, from bowel movements to urination and sex, due to its proximity to important organs in the pelvic area. That makes rectal cancer potentially more dangerous than colon cancer, although it is less common. Moreover, it is more difficult to treat and therefore, needs special expertise to properly diagnose and cure and may recur in 7–21% of people versus about 4–11% of those with colon cancer. There has been a trend over the last few decades of a slightly decreasing incidence of colon cancer in older individuals. However, there is a recent trend of increasing incidences of cancers of the distal colon and rectum in people younger than 50 years, and in general, the cancers in younger individuals are more aggressive. As for the occurrence of both types of cancers in the two sexes, women are more likely to develop right-sided colon cancer, which is more aggressive than left-sided colon cancer, while a larger number of men are more prone to developing rectal cancer than women.

Regardless of these specifics, colon and rectal cancers are combined by the term colorectal cancer. Therefore, in the text, we shall use this combined term for ease and simplicity.

#### 2.2.3. Colorectal Carcinogenesis

CRC generally starts as small, noncancerous (benign) clumps of cells called polyps on the inner lining of the colon or rectum that can become cancerous with time (Figure 2) [47]. There are several forms of polyps, and the chance of a polyp transforming into cancer depends on its type [48]. Adenomatous polyps, or adenomas, sessile serrated polyps (SSP), and traditional serrated adenomas (TSA) have a higher risk of becoming colorectal cancer, while the more common and large (more than 1cm) hyperplastic polyps and inflammatory polyps are benign and not precancerous. CRC most often starts when cells in a polyp begin growing abnormally [49]. As the cancer tumour grows, it invades the deeper layers of the colon or rectal wall. Over time, cancer can grow beyond the colon or rectum and spread into nearby organs—most often, nearby lymph nodes—or can spread into the liver and lungs [50]. The majority of sporadic CRCs (almost 85%) arise from adenomas and have average outcomes. They are characterised by chromosomal instability (CIN), a negative CpG island methylation phenotype (CIMP), microsatellite stability (MSS), and a KRAS-mutated and BRAF wild-type (wt) gene. The other 15% of CRCs originate from serrated adenomas and have been characterised by CIMP-high, microsatellite instability (MSI), and BRAF mutations (Figure 2). They have a good prognosis [49,50].

Genomic instability (chromosomal instability, microsatellite instability, and CpG island methylation phenotypes) plays a critical role in the early stages of CRC development [50,51,52,53]. In the late stages, when CRCs acquire a more aggressive phenotype and become more invasive and metastatic, a critical step is an epithelial-mesenchymal transition (EMT) (Figure 2) [50,51,54,55,56]. In EMT the cells lose their epithelial characteristics, including their polarity and specialized cellular contacts, and acquire a fibroblast-like morphology with a migratory behaviour, allowing them to move to the surrounding tissue. Cell adhesion molecules such as E-cadherin and zona occludens-1, are repressedby some mesenchymal markers, such as vimentin and N-cadherin. Some growth factors, such as the transforming growth factor-β (TGF-β), appeared to be responsible for the induction of EMT. It was also demonstrated that the wingless/int-1 (WNT)/β-catenin signalling pathway as well as the loss of E-cadherin were considered the major effectors of EMT [50,57,58,59,60,61,62] (Figure 2).

#### 2.2.4. Stages of CRCs

Cancer stages describe the extent of cancer development and how far it has spread. The staging of colon cancer is essential in determining the most suitable treatment approaches. Moreover, cancer stages correlate greatly with the cure rate [3,63]. Colorectal cancer staging is done with a number and a letter to indicate the extent of the disease (Table 1) [63]. There are five stages of colon cancer, from zero to four. In stage 0, abnormal cells are found only in the mucosa. These unusual cells may become cancerous. In stage I, the tumour has spread beyond the inner lining but remains within the colon and has not invaded the lymph nodes. In stage II, the colon cancer cells extend through the thick outer muscle layer of the colon without affecting the lymph nodes. Stage II colon cancer is further divided into stages IIA, IIB, and IIC. In stage III, the colon cancer cells migrate outside the colon to one or more lymph nodes. Stage III colon cancer is further divided into stages IIIA, IIIB, and IIIC. In the final stage, IV, the colon cancer cells spread to other parts of the body, such as the liver or lungs. This stage of colon cancer is divided into stages IVA and IVB. Stages 0, I, II, and III are often curable with surgery. However, some patients with stage II and many with stage III are treated with chemotherapy after surgery or receive radiation therapy with chemotherapy either before or after surgery, to increase good treatment outcomes (Table 1). The cure rate is relatively high for early stages CRC patients; 80–95% for stage I and 55–80% for stage II [3,63] while in the last stages the cure rate drops to 5–10% [3,63]. Unfortunately, only 40% of CRC patients are diagnosed at the early stages [3,63]. As for the CRC patients in stage IV, although they cannot be cured, their symptoms, as well as cancer cell growth, can be managed [3].

#### 2.2.5. Current Therapeutic Approaches for CRC

At present, numerous CRC treatment strategies, including surgery, chemotherapy, radiation therapy, targeted therapy and immunotherapy have been employed. They can be divided into local treatments, systemic treatments and combined treatment approaches, and their application depends on the stage of the tumour (Table 1).

Local treatments are those that treat only the tumour without affecting the rest of the body. These treatments are more effective for earlier-stage cancers, but they might also be used in some other situations. The types of local treatments used for colorectal cancer include surgery, ablation and embolization, and radiation. Surgery often called surgical resection, is the most common treatment for CRC [3,64]. It is an effective tool to remove malignant solid tumours at an early stage and when the tumour is well-defined but not very effective in advanced stages [64,65]. In CRC, part of the healthy colon or rectum and nearby lymph nodes can be removed [3,64,65]. The side effects of this surgery include pain and tenderness in the area of operation, along with constipation or diarrhoea [64,65]. Additionally, surgery can increase the risk of death by metastasis in certain cancer patients as a result of the mechanical disruption of the tumour integrity. 

Radiation therapy uses high-energy X-rays to destroy cancer cells by damaging their genetic material that controls how cells grow and divide. It is commonly used for treating rectal cancer because this tumour tends to stay close to its initial place [3,66]. The most common early radiation therapy side effects are fatigue, mild skin reactions, stomach upset, etc. [3,66]. Other side effects might be bloody stools from bleeding through the rectum or blockage of the bowel, sexual problems and infertility in both men and women [66]. Patients have different sensitivity to ionising radiation, which affects the amount of damaged healthy cells. In the more sensitive to radiation damage patients, exposure to radiation therapy can cause sufficient DNA damage to initiate the development of further neoplasms. 

Ablation is often used instead of surgery to destroy small (less than 4 cm) tumours [65]. There are many different types of ablation techniques. Radiofrequency ablation (RFA) uses high-energy radio waves to kill cancer cells. Microwave ablation uses electromagnetic microwaves to create high temperatures that kill cancer quickly, ethanol ablation includes the injection of concentrated alcohol right into the tumour to damage cancer cells, while cryosurgery destroys the tumour by freezing it with a thin metal probe. Possible side effects after the ablation therapy include abdominal (belly) pain, infection in the liver, fever, bleeding into the chest cavity or abdomen, and abnormal liver tests. Embolization is used to treat CRC tumours metastasised in the liver. In an embolization procedure, a substance is injected directly into an artery in the liver to block or reduce the blood flow to the tumour. The possible side effects are similar to those after ablation therapy [65].

Systemic treatments include chemotherapy, targeted therapy, and immunotherapy (Table 1). They are called systemic treatments because they can reach cancer cells throughout the body by the blood flow. Chemotherapy uses drugs to destroy cancer cells, normally by stopping the ability of cancer cells to grow and divide [3,67,68,69,70,71,72]. Depending on the type of colorectal cancer, different types of drugs might be used, applied orally or directly injected into the bloodstream. Currently, several drugs are approved to treat CRC. The current chemotherapy includes both single-agent therapy, which is mainly fluoropyrimidine (5-FU)-based, and multiple-agent regimens containing one or several drugs, including oxaliplatin (OX), irinotecan (IRI), and capecitabine (CAP or XELODA or XEL). Fluoropyrimidine 5-fluorouracil (5-FU), an analogue of thymine and DNA replication inhibitor, has been commonly used in the treatment of CRC. Its intravenous administration, though, was accompanied by big discomfort. The last speed up the process of design of more effective and less expensive oral formulations. Now, these 5-FU formulations are classified into three groups: 5-FU prodrugs such as Tegafur and Capecitabine, 5-FU prodrugs combined with dihydropyrimidine dehydrogenase inhibitor, and 5-FU combined with a dihydropyrimidine dehydrogenase inhibitor [73]. Since the early 2000s, irinotecan (an inhibitor of topoisomerase I) was introduced to the oncological practice, together with oxaliplatin (a DNA cross-linker) [73]. On the other hand, cytotoxic drugs kill proliferating healthy cells such as white blood cells, red blood cells, and platelets. This provoked the simultaneous administration of leucovorin, a vitamin that strengthens the production of blood cells and improves the treatment efficiency during standard CRC chemotherapy [73].

Due to the nonselective cytotoxic mechanism of chemotherapeutic drugs, many strong side effects are reported during and after chemotherapy [3,73]. They include nausea, vomiting, diarrhoea, neuropathy, and mouth sores. Therefore, recently, the attention was set on another systematic treatment for CRC, namely targeted therapy (Table 1). This type of treatment implements monoclonal antibodies, which act against specific molecules on the surface or in the environment of tumour cells [72]. Two drugs in CRC treatment are approved, and they include Bevacizumab, which targets a cancer cell protein called the vascular endothelial growth factor (VEGF), and Cetuximab, which is an epidermal growth factor receptor (EGFR) inhibitor drug used for the treatment of metastatic colorectal cancer [72]. Both medications are based on monoclonal antibodies and are part of the modern therapeutic approaches against CRC.

Cancer immunotherapy (IT) is a novel approach to CRC therapy (Table 1). It uses medicines (immune checkpoint inhibitors) to improve the ability of a person’s immune system to better recognise and destroy cancer cells [6]. Immunotherapy can be used to treat patients with advanced colorectal cancer. It has brought new hopes to cancer patients, as it promises a net improvement in patient survival and quality of life when compared with the standard therapeutic strategies. However, due to immunomodulating features, an optimal time window to combine immune checkpoint inhibitors with other drugs needs to be defined to achieve maximum efficacy while controlling toxicities [70].

In the combined treatment approaches, a combination of different types of treatment is used at the same time or used in a subsequent order, depending on the stage of cancer and other factors. Many people get both chemo- and radiation therapy (called chemoradiation) as their first treatment, which is usually followed by surgery. Additional chemotherapy is then given after surgery, usually for a total of about 6 months. Another option might be to get chemotherapy alone first, followed by chemo plus radiation therapy, then followed by surgery [74,75].

Nonetheless, all techniques have limitations and do not guarantee satisfactory results. The efficacy of conventional chemotherapy, for example, has been decreased by nonspecific distribution and the rapid clearance of anticancer drugs, low efficiency, drug resistance at the cellular and tumour levels, and the considerable toxicity of common anticancer agents when prescribed at higher doses. Hence, new methods are required to further improve their specificity and affectivity [3].

## 3. Nanomaterials: The New Kids on the Block in Colorectal Cancer Treatment

Nanomaterials have attracted worldwide attention due to their capability in improving existing methods for cancer treatment. Some of the most promising therapeutic approaches based on nanomaterials are drug and gene delivery systems, photothermal therapy (PTT), magnetic hyperthermia (MHT), and photodynamic therapy (PDT). The broader applications of nanomaterials in CRC treatment include the customisation of targeted drug delivery systems and advanced treatment modalities [3]. Nanomaterials have several benefits as drug carriers. They increase the water solubility of some hydrophobic drugs and protect drugs dissolved in the bloodstream, improving their pharmacokinetic and pharmacological properties (increasing drug half-lives), target the delivery of drugs in a tissueor cell-specific manner, thereby limiting drug accumulation in the kidneys, liver, spleen, and other non-targeted organs and enhancing their therapeutic efficacy; and deliver a combination of imaging and therapeutic agents for the real-time monitoring of therapeutic efficacy [9,76,77,78,79,80]. Additionally, they can release the drug in a controlled or sustained manner. Stimuli-responsive nanoparticles can also help to lower the toxicity and to control the biodistribution of the drugs [81].

Historically, the development of the nanoparticle design underwent three different stages [77]. In the first stage, the attention has been set on studying the basic surface chemistry of NPs and therefore, they were distinguished by low biocompatibility and high toxicity. The second stage has been focused on the optimisation of the surface chemistry to improve the stability of NPs under biological conditions and to target specific cells. In the third stage, the efforts have been concentrated on the development of stimuli-responsible or the so-called “smart” nanoparticles that can respond to biological (pH, oxidant medium, etc.) or external stimuli (temperature, light, etc.), can target specific cells and improve the biological efficiency [77].

The first discovered nanoparticles used as carriers for drugs and proteins were the liposomes in the 1960s [82]. Since then, many nanomaterials have been developed for drug carriers. In 2016, the FDA approved 51 nanoparticles, while, to date, more than 77 are in clinical trials [83]. Among all the approved materials for nanoparticles, most of them are polymeric and liposomal materials (Figure 3). However, researchers also pay attention to other nanomaterials, such as micelle, metallic, and protein-based materials as drug carriers (Figure 3). Each category of nanomaterials has unique strengths and limitations [81].

Table 2 summarises the advantages and disadvantages of nanoparticles as drug delivery systems. Based on their chemical compositions, nanoparticles and nanostructured materials can be categorised into four types: organic nanomaterials (e.g., micelles, dendrimers, polymersomes, hydrogels, and nanoconjugates); inorganic (e.g., metals, metal oxide, and ceramic nanomaterials); carbon-based (fullerenes, carbon nanofibers, diamonds carbon nanotubes, and graphene); and composite nanostructures. This classification further allows nanoparticle platforms to be broadly categorised as organic, inorganic, and hybrid.

### 3.1. Organic Nanocarriers for Colorectal Cancer Drug Delivery

#### 3.1.1. Polymeric Nanoparticles (PNPs) as One of the Most Widely Used Organic Nanocarriers in Colorectal Cancer Therapy

PNPs are solid, nanosized colloidal particles from biodegradable polymers. Depending on their structure, PNPs can be categorised as nanospheres (matrix type) or nanocapsules (reservoir type). Nanospheres are constructed by a polymeric network, and the bioactive material is dispersed/entrapped into the polymer matrix. Nanocapsules have membrane wall structures with an aqueous or oily core acting as a reservoir to contain the bioactive material. In both cases, the drug can be absorbed on the surface of the sphere or capsule [84,85]. For the preparation of PNPs, natural or synthetic polymers are used. Among the synthetic polymers, the most commonly used are polyethene glycol (PEG), polylactic acid (PLA), polyglycolic acid (PGA), and polycaprolactone (PCL), N-(2-hydroxypropyl) methacrylamide (HPMA) copolymers, polyaspartic acid (PAA), and polyglutamic acid. Among the natural polymers, the most frequently used are chitosan, collagen, dextran, gelatine, albumin, alginate, and heparin [85,86]. All polymers are biodegradable, and in the body, these polymers degrade into monomers, which are removed through normal metabolic pathways. The advantages of PNPs to other nanocarriers used in cancer therapy are their better storage stability, higher drug loading capacity and more controlled drug release than polymeric micelles (PMs) and liposomes, better and controllable physicochemical properties, more homogeneous particle size distribution, and higher drug blood circulation time. All of these characteristics are highly desirable in the context of cancer treatment [85,86,87].

Recently, poly lactic-co-glycolic acid (PLGA) NPs have attracted huge interest because of their well-defined properties in safeguarding drugs’ effective releases. These nanocarriers are developed as transporters for biomolecules (proteins, peptides, and nucleotides), vaccines and small drugs [3,88] and are expansively studied for potential applications in tumour therapy, especially for CRC. Some authors showed that curcumin-loaded PGLA NPs (C-PNPs) demonstrated an advanced cellular uptake in colorectal cancer HT-29 cells compared to pure curcumin, attributable to their sustained release and greater colloidal stability in gastrointestinal fluids and reduced size [89]. These results demonstrated the potential of C-PNPs in the development of an oral targeted drug system in the colon after additional functionalisation with a specific targeting ligand [3,89]. Chitosan, as polymeric NPs, for example, have also been developed as promising carriers for active pharmaceutical components in CRCs [3,90]. Recent data showed that, when the drug SN-38 was encapsulated in poly(D, L-lactide-co-glycolide) NPs and applied to human colorectal adenocarcinoma cell line205 (COLO-205), the detected cellular uptake and cytotoxicity were both approved [91]. For this reason, SN-38-loaded NPs are being investigated as essential drug delivery systems for the treatment of CRC [3,91] (Table 2).

The above-discussed results highlight the polymeric nanoparticles (PNPs) as one of the most widely used organic nanocarriers in cancer therapy. Furthermore, their applications are being broadened constantly, thus promising the effective treatment of different cancers, especially CRC.

#### 3.1.2. Dendrimers as Other Promising Colorectal Cancer Drug Nano-Delivery Systems

Dendrimers are nanosized three-dimensional hyperbranched macromolecules with tree-like arms or branches originating from the central core [85]. The arms have a high number of anionic, neutral, or cationic end groups. Each level of added branches to the core throughout the synthesis process is called a generation. Dendritic macromolecules tend to linearly increase in diameter and adopt a more globular shape with the increasing dendrimer generation [92]. The specific physicochemical properties, along with biodegradable backbones, facilitate the applications of dendrimers as delivery systems for drugs and genes [92,93,94,95]. The drugs and targeting ligands can be loaded onto the cavities in the dendrimer cores through hydrogen bonds, chemical linkages, or hydrophobic interactions. Several dendrimers have been developed for cancer therapeutics: polyamidoamine (PAMAM), PPI (polypropylene imine), PEG (poly(ethylene glycol)), Bis-MPA (2,2-bis(hydroxymethyl) propionic acid), 5-ALA (5-aminolevulinic acid) and TEA (triethanolamine) [93]. Dendrimers have been shown to successfully enhance the efficiency of doxorubicin, a drug commonly used to treat colon cancers [85,96]. Another anticancer drug, cisplatin, in conjugation with a dendrimer, was reported to have enhanced In vitro and *in vivo* activity as compared to free cisplatin after intraperitoneal or intravenous administration into a melanoma mice model having B16F10 tumour cells [85,97]. Zhuo et al. prepared 5-FU–dendrimer conjugates of different generations (0.5–5.5) and showed improved controlled release characteristics for the 5-FU anticancer drug [85,98]. Furthermore, PEGylated dendrimers in conjugation with doxorubicin had improved blood circulation time, reduced toxicity, and resulted in less drug accumulation. Additionally, their subcutaneous administration in a mouse model bearing the highly invasive colorectal cancer C26 cells led to the abolishment of the well-knownDOXunresponsiveness (Doxorubicin) of these tumour cells [85,99]. In another study, PEGylated polyamidoamine (PAMAM) dendrimers conjugated with 5-FU demonstrated a sustained release of the drugs both In vitro and *in vivo* in albino rats [85,100]. Huang W. et al. designed a delivery system for gambogic acid (GA) and other natural anticancer compounds for the treatment of an HT-29 human colon cancer xenograft model based on telodendrimers composed of linear polyethylene glycol (PEG)-blockingdendritic oligomers of cholic acid (CA) and Vitamin E (VE) [3,94]. These nanocarriers showed similar In vitro cytotoxic activity as the free drug against HT-29 human colon cancer cells [3,94] (Table 2). Furthermore, dendrimers can be used as an innovative way of effectively preventing metastatic initiation by the binding and cytotoxic killing of circulating tumour cell CTCs [3,100]. With these promising properties, dendrimers are also called “therapeutic dendrimers” and deserve attention and profound investigation in cancer-targeted therapy [100].

#### 3.1.3. Polymeric Micelles (PMs) as Favourable Organic Nanocarriers in Colon Cancer Therapy

PMs are formed by the spontaneous arrangement of amphiphilic block copolymers in aqueous solutions. They have a hydrophobic core–hydrophilic shell structure thatis suitable for loading hydrophobic drugs into the core, thus improving drug solubility [101]. PM-based carriers are easily prepared and unforced for optimisation. Moreover, PMs can be coupled with targeting ligands to increase accessibility to tumour sites, decrease side effects, and allow controlled drug dissemination over a long period [102,103]. Recently, the development of a pH-responsive copolymer to optimise the delivery of the anticancer drug capecitabine (CAP) using nano-PMs and cyclodextrin (CD) for the treatment of colon cancer was reported. The authors proved that the micelles with their pH sensitivity perfectly target colon cancer, thus delivering the drugs under control to colon tissues with a release above 80% [104]. Therefore, they are considered “smart” nanocarriers for the delivery of anticancer drugs and imaging agents for various therapeutic and diagnostic applications. Noteworthy, several drug-loaded PM formulations have been approved for clinical trials for cancer treatments [105]. Genexol^®^-PM is a paclitaxel (PTX)-loaded PM formulation that is under a phase IV clinical trial for breast cancer treatment (reference number NCT00912639) [105]. Other clinical trials aim to validate it further as a promising chemotherapeutic drug for treating ovarian, breast, lung, cervical and pancreatic cancers.

A large number of preclinical studies on multifunctional PMs are also under way, showing that PM-based drug delivery systems are promising nanomedical platforms for drug delivery and cancer therapy and deserve a lot of attention and further investigation.

#### 3.1.4. Polymeric-Based Nanocarriers as an Emerging Opportunity for Successful Drugs and Nucleic Acids Delivery in Colorectal Cancer

Polymersomes are spherical nanostructures made up of amphiphilic block copolymers, in which a hydrophobic bilayer encloses an aqueous core opposite to the polymeric micelles [93]. Moreover, they represent hollow shell nanoparticles with unique properties to deliver distinct drugs. Polymersome surfaces can be modified with target-specific ligands to bind receptors overexpressed on the surfaces of tumour cells, thus improving the cellular uptake of anticancer molecules [106,107,108,109,110]. Polymersomes have low toxicity, even at high concentrations, because cancer cells internalise them much higher than healthy ones [106,111]. The integration of stimuli-sensitive molecules can also control the drugs released from polymersomes [106,112]. All these benefits make polymersomes widely explored in numerous biomedical applications, including drugs, nucleic acid delivery [106,112,113], gene transfer [114] and diagnostics [115] (Table 2).

Recent results highlighted the properties of two polymersomes: poly(ethylene oxide)-b-poly(γ-methyl-ε-caprolactone) (PEO-PMCL) and vinyl sulfone PEO-PMCL, as harmless bioresorbable polymersomes with high potential for drug delivery. Their binding and release properties were tested in DLD-1 human colon cancer cells, where cisplatin has been encapsulated in them. The unique property of these nanocarriers was the presence of PR_b, which is an α5β1-specific targeting peptide that is specifically expressed in the treated cells. This led to the precise and targeted delivery effectiveness of cisplatin to the colon cancer cells and, in turn, reduced their viability. Interestingly, when these polymersomes with cisplatin were delivered to CACO-2 human epithelial cells, known to express low levels of α5β1 integrin, the cytotoxicity was lowered. This was solid proof that their specificity for drug-targeted delivery was indeed high [116]. These and other results demonstrate that PR-b-functionalised bioresorbable polymersomes serve as a smart approach for minimising the side effects of standard cisplatin chemotherapy for colorectal cancer and deserve a lot of interest and further investigation [117].

#### 3.1.5. Small Extracellular Vesicles (sEVs)—The Trojan Horse for Many Cancers

Small extracellular vesicles (sEVs) have an important role in intercellular communications. sEVs released from different cells exert huge effects on CRC cell proliferation and metastasis. They also serve as biomarkers for CRC diagnosis and prognosis and are potent drug delivery systems in treating CRC patients [118]. EVs are vesicles secreted by the cells into the extracellular space and encapsulate lipids, nucleic acids and proteins. Due to their similarity to natural cell contents, EVs are not recognised by the immune system as foreign substances and, consequently, evade removal by immune cells and reach the target of interest. These unique properties make EVs relevant for drug delivery, expecting to overcome the inefficiency, cytotoxicity, and/or immunogenicity associated with synthetic delivery systems properties. EVs are categorised into three distinct types: microvesicles, exosomes, and apoptotic bodies. Among these three types of EVs, the most used as nanocarriers in cancer treatment are the exosomes [119]. There are studies demonstrating the potential of exosomes to be loaded with anticancer drugs such asDOX and cisplatin and to be delivered to the tumour site. There are signalling-related membrane proteins on the surfaces of exosomes, some of which possess strong immunogenicity and exert antitumour effects by activating immune cells [119,120]. For example, exosomes from dendritic cells (DCs) carry MHC-I, which binds to tumour-derived peptides, and this complex can activate immune cells to exert antitumour effects [119,120,121]. Specific proteins also can be anchored to exosomes through glycosyl-phosphatidyl-inositol (GPI), which further proves that GPI-anchored EGFR nanobodies on the exosomes can bind EGFR-expressing tumour cells. Data show that this binding efficiency was higher than using nanoantibodies directly [119,120,121,122,123].

#### 3.1.6. Liposomes as Colorectal Therapeutics

Liposomes are lipid-based spherical vesicles with an aqueous core enclosed by lipid bilayers [3]. They have single or multiple bilayer membrane assemblies formed from natural or synthetic lipids. Depending on the number of bilayer membranes, liposomes can be unilamellar or multilamellar vesicles and, depending on the size, small or large. Liposomes also vary concerning composition, surface charge, and method of preparation [85]. Due to their smaller size, the ability to incorporate various substances, and phospholipid bilayer in nature, they are considered the most effective drug delivery systems in cells [3,124]. The liposomes are the first nanoparticles used in clinical medicine for drug delivery [3,125] and are still one of the most widely used drug delivery systems for peptides, nucleic acids, and proteins [3,126]. In the mid-1990s, the FDA approved for clinical practice liposome formulations with two drugs: daunorubicin (DaunoXome^®^) and Doxorubicin (Doxil^®^) [3,127]. Later, the FDA approved Marqibo^®^ as a liposomal drug that is a cell cycle-dependent anticancer drug [3,128,129] (Table 2). To overcome drug resistance resulting from liposome-based drugs, they are used as an aptamer to target cancer cells [3,130]. The liposomal delivery system appears to be one of the most promising nanocarriers for CRC treatment. In the last three years, this delivery system has entered extensive preclinical research for CRC therapy [3,131]. Liposome-based nanoproducts, presently under clinical investigations for CRC treatment, are CPX-1, LE-SN38 and Thermodox. CPX-1 (Irinotecan HCl: Floxuridine) successfully ended its phase II clinical trials [132] and recent data show that patients diagnosed with CRC are already getting chemotherapy with oxaliplatin Andirinotecan [132]. The application of the liposome formulation LE-SN38 in HT-29 tumour-bearing mice, on the other hand, was investigated. The results showed that the tumour growth was repressed by 51, 79, and 90% after 10 days of treatment at doses of 10, 20, and 40 mg/kg, correspondingly [133]. Nonetheless, LE-SN38, which was in phase II, was applied at a dose of 35 mg/m^2^ every 21 days for a minimum of two cycles, in patients with metastatic CRC, but did not exert anticancer activity [134]. Thermodox is another liposomal drug candidate in CRC clinical trials. It involves thermally sensitive liposomal doxorubicin as an adjuvant therapy combined with radiofrequency thermal ablation in the treatment of recurrent or refractory colorectal liver metastases [135,136]. And even thought he study comparing Thermodox to radiofrequency thermal ablation monotherapy has been terminated, the results are not yet presented [136]. 

At present, several agents under preclinical investigation have shown auspicious In vitro results with possible applications for CRC, including oxaliplatin-loaded long-circulating liposomes (PEG-liposomal L-oHP) [137], liposomal curcumin [138] and doxorubicin-encapsulated liposomes [139]. 

These data highlight the potential of liposomes as drug carriers in colorectal carcinoma cells and further deliver promises for decreasing the adverse effects of standard chemotherapy.

### 3.2. Colorectal Cancer Drug Delivery Based on Inorganic Nanocarriers

#### 3.2.1. Mesoporous Silica Nanoparticles (MSNs) as Carriers of Large Amounts of Biomolecules

MSNs belong to silica (SiO_2_) class materials that have gained huge interest for drug delivery due to their beehive-like porous structure that allows the packing of large amounts of bioactive molecules. Other important features of MSNs are the adjustable sizes of cavities in the range of 50–300 and 2–6 nm, respectively, the low level of toxicity, easy endocytosis, and resistance to heat and variable pH [3,140,141]. The MSN–protamine hybrid system (MSN-PRM) has been designed for a more selective release of the drugs to cancer cells and was activated with specific enzymes to trigger anticancer activity [3,142]. The conjugation of MSNs with hyaluronic acid has been shown to significantly increase the amount of DOX loading into HA-MSNs compared to bare MSNs [3,143] and to increase the cellular uptake of DOX-HA-MSN conjugate, which was reflected by the enhanced cytotoxicity of the human colon carcinoma cell line (HCT-116 cell line) [3,144]. In another study, MSNs were functionalised with polyethyleneimine-polyethene glycol (PEI-PEG) and polyethylene glycol (PEG), which also increased the amount of loading of epirubicin hydrochloride (EPI) and exhibited improved antitumour activity [3,145] (Table 2).

In CRC treatment approaches, silica nanoparticles were used together with a photosensitizer to destroy the CRC cells [146]. These silica-based nanoshells have been designed to enclose photosensitizer molecules in such a way to be easily taken up by the CRC tumour cells. Upon exposure of these cells to light, the photosensitizers are activated thus releasing reactive oxygen molecules to kill the cancer cells [3,147]. Currently, this technique is adopted in cancer treatment in numerous clinical trials. Thus, nanoparticles may also be applied for the molecular imaging of cancer cells, followed by earlier diagnosis and targeted drug delivery.

#### 3.2.2. Metallic and Magnetic Nanomaterials as Photosensitizers in Colorectal Cancer Treatment

Metallic and magnetic nanomaterials have some specific optical, magnetic, and photothermal properties that make these types of nanomaterials useful for different biomedical applications. Iron oxide is a prime example of a metallic nanomaterial that can be used in different ways. It has super-magnetic properties that allow it to be used both for imaging techniques and for the targeted delivery of drugs and molecules. Metallic nanoparticles can be conjugated with other types of nanomaterials and can be combined with other types of therapy, such as photothermal therapy (PTT). Iron oxide NPs have excellent biodegradability *in vivo* because after dissolution in the human body, the iron ions can be adjusted physiologically [3,148]. Recent reports have appeared that show that smart multifunctional magnetic nanovesicles encapsulating the antibody-targeting peptide AP1 (MPVA AP1) are a promising anticancer drug [3,149]. These nanoformulations did not exhibit a cytotoxic effect in L929 fibroblasts but demonstrated outstanding selectivity and targeting toward CRC cells (CT26-IL4R). No leakage of encapsulated drugs in the absence of magnetic field stimulation has been revealed. Moreover, nanovesicles loaded with doxorubicin burst upon treatment with a high-frequency magnetic field, which caused a fast, accurate, and controlled drug release. Therefore, smart magnetic nanovesicles like MPVA-AP1 have a notable capability for delivering specified doses and precisely controlled releases in antitumour applications [149]. Iron oxide NPs enhance the effect of hyperthermia and areextremely beneficial in the diagnosis of CRC [150]. For example, polylactide-co-glycolic acid (PLGA) nanoparticles loaded with 5-fluorouracil (5-FU) and iron oxide demonstrated greater DNA damage in an HT-29 colon tumour cell line as compared with hyperthermia [150]. Other studies have shown that paclitaxel (PTX) and the super-paramagnetic iron oxide (SPIO), encapsulated inside the core of PEAL Ca micelles [151], have been released from micelles slower at neutral pH and faster at a pH of 5.0. Cell culture studies have also shown that PTX-SPIO-PEALCa is successfully absorbed by CRCLoVo cells, while PTX is internalised by lysosomal cells. Additionally, the successful inhibition of CRC LoVo cell growth has been verified. Hence, micelles promise a great role in MRI visible drug release methods for CRC treatment [151] (Table 2).

#### 3.2.3. Carbon-Based Nanomaterials

The family of carbon-based nanomaterials includes fullerenes, carbon nanotubes, graphene and its derivatives, nanodiamonds, and carbon-based quantum dots [152]. This diverse family of nanomaterials has unique physicochemical features like excellent mechanical, electrical, thermal, optical, and chemical properties. Therefore, they gather huge interest and are under research in diverse areas, including biomedical applications as carriers of various types of therapeutic molecules for disease treatment and tissue repair and the imaging of cells and tissues. Moreover, their antibacterial and anti-inflammatory properties are also extensively studied [152].

Carbon nanotubes (CNTs) are broadly used types of carbon nanoparticles in biomedical research. They are cylindrical, with rolled graphene sheets, with a diameter of fewer than 1 μm and a few nanometres in length [3,153]. Their high surface areas, needle-like structures, heat conductivity, and chemical stability enable their use in immunotherapy, diagnosis, gene therapy, and a carrier in different drug delivery systems [3,154]. Several antitumour-drug-carriers-strategies based on CNTs have been developed. In one of them, the single-walled carbon nanotubes (SWCNTs) conjugated with a synthetic polyampholyte for the delivery of paclitaxel demonstrated greater anticancer effects in Caco-2 and HT-29 cells compared to paclitaxel alone [3,155]. A similar effect was observed using Eudragit^®^–irinotecan-loaded CNTs [3,156]. Additionally, oxaliplatin and mitomycin C-coated CNTs, induced by infrared light rays, have shown a considerably higher drug delivery and localisation in cancer colon cell lines [3,157]. Some data show that SWCNTs modified with TRAIL (a ligand that binds to specific receptors and causes the apoptosis process in cancer) increase cell death ten times in comparison with the mono-delivery of TRAIL in carcinoma cells lines [3,158].

Recent progress in nanotechnology and biotechnology has contributed to the development of precisely targeted cancer therapies based on nanomaterials. Carbon nanomaterials and graphene play a pivotal role in this due to their high variability, chemical stability, and unique characteristics. However, to enable drug delivery platforms, the major emphasis should lie in the investigation of the molecular mechanisms of nanoparticle delivery to the cell. These pathways are crucial in allowing the effect of the nanoformulations on cellular compartments and cellular fate in general. Moreover, a detailed understanding of the pharmacologic and toxicological properties of these nanomaterials, and a well-adjusted and detailed assessment of their risks and benefits to human health, have beenunder broad investigation recently. 

We provided a schematic illustration of some of the pathways for NP delivery to the cells and highlighted the implications on the cellular compartments and their fates in Figure 4. The well-studied mechanism of nanoparticle delivery to the cell involves the following steps: NPs get into cells in different ways, mainly through diffusion, endocytosis, and/or binding to receptors, and induce reactive oxygen species (ROS) generation, LDH and MDA increase, and cytokines release, which, in turn, can cause oxidative stress, loss in cell functionality, pro-inflammatory responses, and mitochondrial damage. The uptake of graphene into the nucleus may cause DNA-stranded breaks and the induction of gene expression via the activation of transcription factors, cell death, and genotoxicity. 

## 4. Graphene-Based Nanomaterials—An Everlasting Source of Innovations in Biomedicine

### 4.1. Graphene Oxide Synthesis and Structure, Types, Properties and Applications in Anticancer Treatment

Graphene-based nanomaterials fall within the scope of carbon nanomaterials, and due to their broad applications, we have summarised and discussed their advantages and disadvantages separately in the text below.

The discovery of two-dimensional graphene by Novoselov et al. in 2004 initiated a new era in the research on nanometre-sized materials [159,160]. Graphene oxide (GO) is the oxidised, water-soluble form of graphene and the most recommended graphene derivative for biomedical use [161]. In 2008, Liu et al. demonstrated, for the first time, the potential of graphene oxide conjugated with polyethylene glycol to serve as a drug carrier for water-insoluble, aromatic anticancer drugs (SN38) [7,15]. Since then, the research on graphene oxide-based materials has launchednew perspectives to improve the therapeutic efficiency of cancer treatments [7]. Several graphene oxide-based nanomaterials have been under extensive research recently, and they include pristine GO, nanographene oxide, reduced GO (rGO), and differently chemically modified graphene oxide nanocomposites. 

In this part of the review, we shall briefly present the development and application of these types of graphene oxide-based nanomaterials and shall then focus on their applications in colorectal carcinoma cells treatment.

#### 4.1.1. Pristine Graphene Oxide (GO)

Pristine GO is a two-dimensional (2D) material consisting of carbon atoms organised into a hexagonal, honeycomb structure and large quantities of oxygen-containing functional groups on its surface and edges. The unique combination of extraordinary properties of GOare superior to many of the existing materials and offer a fascinating platform for the development of next-generation technologies in many fields [7,162]. The extraordinary properties of GO originate from the one-atom-thick planar sheet structure; aromatic (sp^2^) and aliphatic (sp^3^) domains; *π*–*π* and/or *n*–*π* orbital interactionsand abundant oxygen-containing groups (hydroxyl, epoxy, carboxyl, and carbonyl groups), which can easily bond with other biomolecules and polymers [163]. These physicochemical properties are the reason why it is generally accepted that GO has multifunctional applications. GO has deformations in the planar structure due to the high density of sp^3^ carbons, resulting from the insertion of functional groups containing oxygen, making it more soluble in water [164]. The exact structure of GO is still under debate because it strongly depends on the synthesis method and the reducing agents [165]. 

Over the years, several models have been proposed for the structure of graphene oxide. In the first model proposed by Hofmann in 1939, the carbon skeleton was composed of sp^2^ hybridisation and epoxide groups distributed throughout the graphene plane, giving a C/O ratio of 2 [164]. Ruess introduced hydroxyl groups into the model in 1947 [165] and suggested a carbon structure composed of cyclohexane repeating units of sp^3^ hybridisation. The most recognised model is that of Lerf and Klinowski called the L–K model. In 1998 [166], they proposed a model of a flat carbon lattice structure with an unoxidized benzene ring region and an oxidised aliphatic six-membered ring region. The relative sizes of these two regions depending on the degree of oxidation. Hydroxyl and epoxy groups are randomly distributed on the graphene oxide single layer, while the carboxyl and carbonyl groups are situated at the edge of the single layer. The authors also pointed out that the carbon atoms attached to the OH groups slightly distorted their tetrahedral structures, resulting in a slight folding of the layer. However, this model ignored the influence of raw graphene and the synthesis methods. Erickson et al. [167] found that graphene oxide had hole defects due to over oxidation and lamellar peeling. In recent years, other models have been proposed, such as the dynamic structure model (DSM) and the binary structure model [168].

The DSM model explained the acidity of the aqueous GO solutions [169,170,171,172]. The recently proposed binary structure model of Liu et al. [172,173,174] proposed C=O bonds on the edge and plane of GO, which partly confirmed the models proposed previously.

Extensive research has been conducted on the preparation and properties of GO, and the different methods of preparation of graphene oxide are well-described and could be found in the papers of Liu et al., Tais Monteiro Magne et al., and many others [7,169].

There are many classic methods for synthesising GO, including the Brodie method [175], Hummer’s [176], and the Marcano method [177]. Although graphene was isolated in 2004 by Geim and Novoselov, the oxidised form of graphene - graphene oxide - has been known since the 19th century [159,172]. In 1960, Brodie [175] prepared graphite oxide for the first time by exposing graphite to strong acids and obtained a graphite oxide suspension that was multilayer stacked graphene sheets densely covered with hydroxyl and epoxide groups. Later, Staudenmaier modified the method to increase the oxidation process by adding small amounts of chloride and sulfuric acid to increase the acidity [178]. Nowadays, the most widely used method for graphene oxide synthesis is the Hummer’s method, developed in 1958, using potassium permanganate (KMnO_4_), sodium nitrate (NaNO_3_) and sulfuric acid (H_2_SO_4_) [176]. Sulfuric acid aids graphite oxidation with KMnO_4_, as it acidifies the pH. Additionally, the Mn_2_O_7_ formed by the reaction of sulfuric acid and KMnO_4_ possesses a strong oxidation ability, which plays a crucial role in forming graphene oxide. Hummer’s method was adapted from Brodie’s graphite oxide synthesis. In general, the typical synthesis of graphene oxide goes through two steps: oxidation of graphite powder with various oxidants in an acidic environment and exfoliation, in which GO is obtained by ultrasonication or mechanical stirring [174].

#### 4.1.2. Nanographene Oxide (nGO)

Nano-GO (nGO) is a GO derivate with the same 2D layer structure of sp^2^-bonded carbon atoms in a hexagonal conformation and sp^3^ domains with carbon atoms linked to oxygen functional groups but with nanometre dimensions. It is obtained from the GO by ultrasonication aiming to convert the micrometric lateral dimension of GO sheets to a nanometric size under 100 nm [77]. The smaller size of nGO facilitates the penetration of nanoparticles into cells.

#### 4.1.3. Reduced Graphene Oxide (rGO)

rGO is another derivative of GO with a much lesser number of oxygen functional groups than GO. Reduction of the oxygen-containing functional groups on the surface of GO and obtaining reduced GO can be performed by chemical, thermal, or electrical methods. The chemical reduction method prevails over nonchemical methods for rGO synthesis, and it is the most promising method for the large-scale production of rGO. Different reducing agents lead to various carbon-to-oxygen ratios and, correspondingly, to different chemical compositions in rGO. Upon the reduction process, most of the oxygen groups and sp^3^ carbon are eliminated to generate a more graphene-like material with improved properties [179]. Indeed, the chemically reduced graphene sheet undergoes incomplete reduction, leaving behind a few oxygen-containing functional groups [162].

Chemically reduced graphene oxide looks similar to pristine graphene in terms of its electrical, thermal, and mechanical properties. The most common chemical reagents used for the direct or indirect reduction of GO are hydrazine and hydrazine hydrate. Hydrazine-based reduction increases the presence of sp^2^ domains on graphene sheets. Other chemical reducing agents are sodium borohydride, hydroquinone, gaseous hydrogen, strong alkaline solutions, and ascorbic acid [162]. The chemical reduction of GO, though, leads to impurities in the final product, which is recognised in the field as a disadvantage. [162]. Additionally, many of the commercially available chemical reducing agents are toxic, while, for the biomedical application of rGO, nontoxic green chemicals for the reduction process are preferred. Green tea extracts, resveratrol, and chitosan, for example, are eco-friendly reducing agents, which are cheap, biocompatible, and easily available [179]. Fungal extracts can also be applied as reducing and stabilising agents for the reduction of GO because of nontoxicity, availability and absence of toxic waste from the process of synthesis. Compared to GO, rGO has a less aromatic character, limited solubility in an aqueous medium, and tends to aggregate through π–π interactions. However, rGO exhibits much greater NIR absorption than pristine GO, and it is easier to be used as a photothermal agent for cancer therapy [163].

#### 4.1.4. Graphene Oxide Nanocomposites

We already highlighted GO’s exceptional mechanical, chemical, and thermal properties. They make graphene oxide an excellent candidate for composite applications. GO has a large and easy-to-modify surface due to oxygen-containing functional groups (epoxide hydroxyl; carboxyl; and hydroxy (–O–, –COOH, and –OH), which allow easy changes of the GO surface, thus providing attachment sites to various molecules [7] as well as synthetic polymers such asPEG, poly-l-lysine (PLL), polyvinyl alcohol (PVA), and Pluronic F127 (PF127) and natural polymers such asCS, sodium alginate (SA), dextran (DEX), l-cysteine and gelatine [7]. The proper functionalization of graphene oxide prevents agglomeration during the reduction of graphene oxide and preserves its inherent properties. The functionalization of graphene oxide results in new properties, such as good dispersion and biocompatibility, excellent mechanical properties, electrical properties, and thermal properties, which subsequently extend its functions and applications in biomedicine [172]. These types of functionalization are achieved by covalent and noncovalent modifications [7,180]. Sulfonylation and acylation are the most widely applied covalent modifications of GO [7,181]. On the one hand, though, these modifications destroy the unique structure of GO by the hybridisation of sp^2^ carbon atoms of the π-network into the sp^3^ configuration but, on the other hand, provide it with stability under physiological conditions. The noncovalent modification of GO by Van der Waals forces, electrostatic interactions, hydrogen bonding, and π–π stacking interactions do not affect the unique structure of GO but increase its instability [7,182].

### 4.2. Graphene Oxide in CRC Treatment Strategies

Nanomaterials based on GO are seriously examined for anti-cancer treatment strategies like chemo, photothermal (PTT), photodynamic (PDT)and gene therapy. They are also preferred in combinational therapeutic strategies, such as chemo/phototherapy and photothermal/immunotherapy. In this section, we summarized the latest studies on GO-based nanoparticles for CRC therapy and categorized them into several subsections depending on the treatment modality used (Table 4) [179].

#### 4.2.1. GO as Drug Delivery System in CRC Treatment

The most commonly used chemotherapeutic drugs are Doxorubicin (DOX), cisplatin (CDDP), Paclitaxel (PTX), and Camptothecin (CPT), Mitoxantrone (MTX), Vinblastine (VIN), Vincristine (VCR), and 5-Fluorouracil (5-FU) [183,184]. They inhibit the growth rates and metabolic functions of different cancer cells. The most common chemotherapeutic drugs used in CRC therapy are Avastin, Oxaliplatin, Capecitabine, 5-FU (fluorouracil), Leucovorin, and others [184]. However, many complications are reported in mono-drug chemotherapies due to their cytotoxicity towards normal cells, drug resistance, lack of targeting delivery and controlled release, which limits their application in cancer therapy.

In the past few years, graphene oxide-based nanomaterials have been introduced as a new type of drug delivery system with the potential to overcome these limitations and deliver chemotherapeutic drugs with minimum side effects due to their exceptional properties, such as a high loading efficiency, relatively low toxicity, good surface properties and excellent water solubility. Moreover, the hydrophobic features of GO and its high surface area allow various modifications and drug loading by noncovalent bonds, such as π–π stacking and hydrophobic contacts. Interestingly, the hydrophilic groups such as hydroxyl, carboxyl, and epoxy further offer reactive sites to connect functional molecules to improve anticancer efficiency [7,163]. Indeed, the surface area of a single GO layer is 2600 m^2^ g^−1^, which is much higher than other nanomaterials and permits a much higher drug loading capacity [7]. A drug loading value of GO has been estimated as 235 weight percentage (wt%), much higher than other nanomaterials [7]. In 2008, Liu et al. demonstrated the potential of PEG-modified GO to bind the active metabolite 7-ethyl-10-hydroxycampothecin (SN38) via Van der Waals forces for the treatment of CRC [15]. Some other drugs such as paclitaxel (PTX) were also loaded onto GO-PEG through π–π stacking and hydrophobic interactions. The GO-PEG-PTX nanocomposite has shown a higher cytotoxicity effect and excellent bioavailability than free PTX [7]. In other research, graphene oxide (GO) was functionalized with hydrophilic and biocompatible Pluronic F38 (F38), Tween 80 (T80) and maltodextrin (MD) for the loading and delivery of poorly water-soluble ellagic acid (EA), an antioxidant and anticancer drug. The release of EA from these nanocarriers was studied in water (neutral pH) and buffer solutions of pH 4 and 10, at 37 °C. The functionalized GO with EA showed a better capability to kill human breast carcinoma cells (MCF7) and human colon adenocarcinoma cells (HT29) than that of free EA dissolved in DMSO. Additionally, the loading of EA onto the functionalized GO did not hamper its antioxidant activity. These results confirmed that the functionalized GOs were suitable and played the role of successful nanocarriers of drugs for colorectal cancer therapy [161] (Table 3). In the work of Lina A. Al-Ani, (2020), curcumin AuNP-reduced graphene oxide nanocomposite (CAG) was developed, and the impact on cellular membrane integrity and the possible redox changes were studied In vitro on drug-responsive and drug-resistant colon cancer cells: HT-29 and SW-948 cell lines [185]. The results proved the killing of both types of colon cancer cells (HT-29 and SW-948) by apoptosis, while the measured redox parameters revealed a time-dependent increase in ROS accompanied by regressive GSH and SOD levels. This was a successful demonstration of the realisation of graphene oxide as a platform for curcumin intracellular delivery and cytotoxicity for colon cancer cells regardless of their cancer properties [185]. Other examples include the synthesis of dual drug-encapsulated chitosan/rGO nanocomposites by entrapping 5-fluorouracil (5-FU) and curcumin in chitosan/sodium tripolyphosphate gel in the presence of rGO nanosheets. The authors registered synergistic cytotoxicity toward HT-29 colon cancer cells compared with single-drug therapy [179,186] (Table 3).

Colon cancer drug delivery has been a challenge since the technologies emerged. Stimuli-responsive graphene oxide-based nanomaterials were further designed considering the unique characteristics of the tumour microenvironment and different approaches to trigger the release of drugs from GO carriers at the site of the tumour, including internal cellular changes in the pH and redox, and exogenous stimuli such as near-infrared (NIR) light, ultrasound, magnetic, and electric fields were developed [7]. This is one of the key points in the design of a successful drug carrier with controllable and complete drug discharge at the tumour site [7]. Several authors [187,188,189] reported the design of GO-PEG nanocarriers for DOX delivery that was sensitive to the pH. These results confirmed a more efficient and targeted release of DOX in the acidic, i.e., tumour environment when compared to normal tissue. The reason for this was GO sensitivity to low pH sensitivity, at which the hydrogen bonds between the drug and GO were weakened. Other authors reported on graphene-based carriers with pH-sensitive polymers to accelerate the release of drugs in the acidic cancer environment. For example, Bai et al. developed a pH-sensitive GO/polyvinyl alcohol hydrogel for the loading and release of vitamin B12 at the physiological pH and demonstrated pH and salt-dependent drug release [190]. The examples do not end here. Other data discuss the properties of mesoporous silica (MS)-modified GO as a pH-responsible carrier for anticancer drug delivery—more specifically, for epithelial neuroblastoma cells. In these studies, the acidic sensitivity of nanomaterials was tested in phosphate-buffered saline at pH 7.4 and 5.5, at which the release fraction was 55.2% and 69.0%, respectively [191].

Fiorica et al. 2017 revealed that the release of Irinotecan, a common chemotherapeutic for colorectal cancer, loaded on GOmodified with hyaluronic acid/polyaspartamide was influenced both by the pH of the external medium and NIR (near-infrared) light irradiation [192]. Other authors reported on the development of a safe and efficient nanocomposite hydrogel for colon cancer drug delivery that was pH-sensitive, as well as biocompatible [193] (Table 3). These features were achieved by azoaromatic crosslinks, as well as by the modification of graphene oxide (GO) with polyvinyl alcohol (PVA). In these nanocomposite GO gels, curcumin (CUR), an anticancer drug, was encapsulated successfully through freezing and thawing. The experimentations under conditions mimicking stomach to colon transit for the drug release revealed that the drug was protected from the adverse physiological conditions of the stomach and the small intestines. This proved that GO–N=N–GO/PVA composite hydrogels protected CUR and improved the colon-targeting ability and residence time. This proved the potential of CUR-loaded GO–N=N–GO/PVA composite hydrogels to potentially provide a theoretical basis for the treatment of colon cancer.

#### 4.2.2. Strategies for GO-Based Targeted Therapeutic Approaches for CRC

Targeted drug delivery is a concept that has developed rapidly in recent years. It has become one of the most popular strategies for cancer treatment since it can substantially increase the treatment efficiency with fewer side effects than conventional methods. A recent development in GO-based therapeutic approaches for fighting CRC focused on GO-based targeted therapy against different cancer moieties to increase drug uptake in tumour tissues and minimise it in normal. Tumour targeting is realised by bonding drug carriers with specific ligands (antibodies, antibody fragments, aptamers and other small molecules) and recognising specific molecules on the surfaces of the target cancer cells, such as folic acid, transferrin, vascular endothelial and epidermal growth factors, which are commonly upregulated in cancer cells (Figure 5) [7,194,195,196,197].

The colorectal carcinoma cells overexpress folate receptors [197,198,199]. There is evidence that folate ligand-based nanocarriers are selectively internalised into the tumour cell by folate receptor-mediated uptake [200]. Additionally, an attractive targeting ligand for the effective treatment of colon cancer is wheat germ agglutinin (WGA), demonstrating a greater affinity toward the colon cancer cells as compared to the other plant lectins [197]. Other commonly used ligands for tumour targeting arehyaluronic acid, folic acid, transferrin, vascular endothelial and epidermal growth factors (Figure 5). Jiang et al. for example used hyaluronic acid (HA) as a targeting ligand to specifically deliver photosensitizer chlorine e6 (Ce6) to CD44 overexpressing HCT-116, and HT29 colon cancer cells for PDT with NIR irradiation [200]. The authors observed increased cellular uptake of the newly fabricated rGO-based nanoplatform (rGO-PDA@MS/Ce6/HA), leading to a more significant destruction of the target cancer cells as compared to the nanoparticles without a targeting ligand (rGO-PDA@MS/Ce6) [200]. Some of those ligands, however, are not effective when used in a single anticancer therapy, therefore, combination therapy is needed. For example, the combination of the anticancer drug with Bevacizumab (Avastin, Genentech), an anti-VEGF monoclonal antibody, improved the clinical benefits in a patient with metastatic CRC [197]. The novel targeted drug delivery carriers facilitate the effective intracellular access and accumulation of the drug in tumour cells.

#### 4.2.3. Photothermal Therapy (PTT) and Photodynamic Therapy (PDT) Are Promising Adjuvant Cancer Therapies

Photothermal (PTT) and photodynamic therapy (PDT) are new approaches to the development of modern cancer cures, both characterised by advantages such as remote controllability and spatiotemporal selectivity without increasing harmfulness [7,200,201]. Recently, in the field of CRC therapy, the focus has been set on mono- and combined PTT and PDT approach involving graphene oxide-based nanomaterials (Table 4).

PTT is a technique that converts light energy into heat by the activation of photosensitising agents with near-infrared (NIR) irradiation, thus increasing the local temperature and inducing cancer cell death. Graphene oxide-based nanomaterials possess strong absorption in the NIR region due to the specific delocalised electron arrangement. Therefore, they can act as photothermal sensitizers [7,202]. Many studies are focused on studying GO as a PTT agent for cancer therapy [7,203]. Robison et al. first demonstrated the much higher NIR absorption rate (six-fold) of chemically reduced GO than pristine GO and the high effectiveness of rGO as a PTT agent over GO [182,203]. In a study by Abdolahad M et al., green tea-reduced GO resulted in a 20% higher photothermal destruction of the high metastatic colorectal carcinoma cells of SW48 than that of low metastatic HT29 cells [179,204] (Table 3). Other authors have further proved that, when PTT was combined with folic acid as a targeting ligand by conjugating rGO nanomaterials with the indoleamine-2,3-dioxygenase (IDO) inhibitor, the combinedimmunotherapy led to synergistic antitumour immunity by IDO and programmed cell deathligand 1 (PD-L1) in CT26 cells. Furthermore, the authors showed that laser irradiation of the as-built nanocomposites directly destroyed tumour cells due to the PTT and activated antitumour immune response synergistically by IDO inhibition, as well as the PD-L1 blockade in CT26 colon cancer cells. These results proved that synergistic antitumour immunity can effectively be induced by targeting multiple antitumour immune pathways [179,205]. The data showed that hyaluronic acid/polyaspartamide-based double-network nanogels, containing GO loaded with irinotecan for the potential treatment of colorectal carcinoma (HCT 116), revealed that nanogels were uptaken by the cancer cells and, in the presence of the antitumour drug, produced a synergistic hyperthermic/cytotoxic effect. Moreover, the authors demonstrated that it was possible to conduct the thermal ablation of solid tumours after the intra-tumoral administration of the nanogels [192].

PDT has been already approved by the Food and Drug Administration (FDA) as an effective cancer therapy [7,205,206]. It is another form of phototherapy involving specific light irradiation and a photosensitizer (PS), but unlike PTT in PDT, it produces reactive oxygen species (ROS), through which cancer cell death is achieved [179]. GO was found to act both as a photothermal agent for PTT and a photosensitizer for PDT, thus demonstrating its potential for synergistic phototherapy [179].

Some authors have loaded chitosan-reduced graphene oxide (chit-rGO) with IR820 dye and doxorubicin (DOX) and have developed a nanotherapeutic platform integrating chemotherapy, PTT and PDT. Theauthors have studied the ability of the as-prepared chit-rGO-IR820-DOX nanoplatform to generate singlet oxygen and heat under NIR irradiation with 785-nm wave length as well as its In vitro anticancer activity against murine colon carcinoma cells (C26). The results have confirmed the synergistic anticancer activity of the fabricated nanosystem against C26 cancer cells [207]. Other authors, such as Jiang et al., have developed a tumour-targeting photothermal heating-responsible nanoplatform based on reduced graphene oxide/mesoporous silica/hyaluronic acid nanocomposite for enhanced PDT [200]. The proposed nanoplatform specifically delivers chlorine e6 (a photosensitizer) to CD-44 overexpressing colon cancer cells (HCT-116 and HT29) for PDT with a NIR irradiation response. The combination of excellent NIR photothermal conversion ability and controllable Ce6 (photosensitizer chlorin e6) release results in an enhanced singlet oxygen generation, which leads to elicit ablation of the target colon cancer cells [200].

### 4.3. Direct Killing Effect of Graphene Oxide

The current therapeutic applications of GO and its derivates toward CRC are mainly focused on the development of targeted drug delivery and PTT systems (Table 4). GO, however, not only functions as an effective drug carrier and PS but can also exert a direct killing effect on cancer cells by inducing apoptosis and the generation of reactive oxygen species. There is evidence for the In vitro and *in vivo* toxicity of GO indifferent cancer cells and extremely low toxicity innormal cells. Bera S. et al. showed that GO is more toxic to human colon cancer cells (HCT116) compared to human embryonic kidney epithelial cells (HEK293) [208] (Table 3). In HCT116 cells, GO disrupted the microtubule assembly, which led to cell cycle arrest in the S phase and induced apoptosis by generating reactive oxygen species (ROS) in a dose and time-dependent manner. 

In our studies, we evaluated the cytotoxic and genotoxic effects of pristine and aminated GO particles on different cancer cells, including colon C26 cancer cells, lung cancer A549 cells, and hepatocellular HepG2 cancer cells. We found that the mechanisms of GO cytotoxicity were different and depended on the cell and GO particle types. In Colon26 aminated GO induced ROS production and blocked the cells in the G0–G1 phase of the cell cycle, while in lung cancer cells (A549) aminated GO induced greater DNA damage, lower ROS production, and blockage of the cell cycle in the G2–M phases [209,210,211].

Another report suggested that GO inhibited the migration and invasion of various cancer cells by inhibiting the activities of electron transport chain (ETC) complexes presented in mitochondria, resulting in reduced ATP synthesis, inhibiting F-actin cytoskeleton assembly [212]. Some data showed the potential of GO to provoke autophagy in cancer cells. GO itself was shown to induce Toll-like receptor (TLRs) responses and autophagy in cancer cells, thus conferring antitumour effects in mice. The possible underlying mechanism was in the phagocytosis of GO by CT26 colon cancer cells, simultaneously triggering autophagy, as well as TLR-4 and TLR-9 signalling cascades. The authors dissected the crosstalk between the TLRs and autophagy pathways and uncovered that the GO-activated autophagy was regulated through the myeloid differentiation primary response gene 88 (MyD88) and TNF receptor-associated factor 6 (TRAF6)-associated TLR-4/9 signalling pathways. The injection of GO alone into immunocompetent mice bearing CT26 colon tumours not only suppressed the tumour progression but also enhanced cell death, autophagy, and immune responses within the tumour bed [213] (Table 3 and Table 4).

### 4.4. GO as a Chemosensitiser

Recently, GO also emerged as a chemosensitizer, making cancer cells more susceptible to chemotherapeutic agents. Recent findings by Zhu et al. 2017 [214] showed that GO treatment at sublethal concentrations impaired the general cellular priming state, including disorders of the plasma membrane and cytoskeleton construction in J774A.1 macrophages and A549 lung cancer cells without incurring significant cell death but, at the same time, dampening several biological processes. The underlying mechanism revealed that GO interacted with the integrin receptor on the plasma membrane and consequently activated the integrin−FAK−Rho−ROCK pathway and suppressed the expression of integrin, resulting in a compromised cell membrane and cytoskeleton and a subsequent cellular priming state. Based on this mechanism, it was concluded that the efficacy of chemotherapeutic agents (e.g., doxorubicin and cisplatin) could be enhanced by the GO pre-treatment of cancer cells. These results unveiled the feature of GO in cancer therapeutics to sensitise cancer cells to chemotherapeutic agents by undermining the resistance capability of tumour cells against chemotherapeutic agents, at least partially, by compromising plasma membrane and cytoskeleton meshwork [214].

GO’s the chemosensitizing activity was also demonstrated based on the enhanced anticancer activity. GO acted synergistically with the chemotherapeutic agents and perturbed the plasma membrane, disrupted the cytoskeleton meshwork, diverted the autophagy flux, enhanced nuclear imports, and induced necrosis [208]. There were indications for both direct and indirect interactions of GO with F-actin cytoskeleton assembly, which resulted in potential anticancer activity. It has been shown that PEG-grafted GO entered the cell and interacted with the F-actin cytoskeleton, inducing cell cycle alterations, apoptosis, and oxidative stress [208].

GO was also proved to induce autophagy in CT26 colon cancer cells, thus conferring its antitumour effects. When GO and the chemotherapy drug cisplatin (CDDP) was used alone, they induced autophagy, but the degree of CT26 cell death was low. When GO was combined with CDDP (GO/CDDP), it potentiated the CT26 cell killing via necrosis by inducing the nuclear import of CDDP and the autophagy marker LC3. GO/CDDP diverted the LC3 flux in the early phase of autophagy, resulting in LC3 trafficking towards the nucleus in an importin-α/β-dependent manner, which concurred with the CDDP nuclear import and necrosis. The intra-tumoral injection of GO/CDDP into mice bearing CT26 colon tumours potentiated immune cell infiltration and promoted cell death, autophagy, and HMGB1 release, thereby synergistically augmenting the antitumour effects [213].

## 5. Conclusions and Perspectives

Nanomaterials hold a vast sea of possibilities in cancer therapies, though many are yet under extensive studies. Data for their formulations, cyto- and genotoxicity, as well as molecular mechanisms of action, are accumulating rapidly. To benefit from them, the anticancer research needs filtering and analysis, as well as a precise definition of the type of tumours successfully targeted by nanomaterials concerning their formulations and modifications.

Nanoparticle drug delivery systems have the potential to improve the current cancer therapies because of their ability to overcome multiple biological barriers and release a therapeutic drug in the optimal dosage range. They can also improve the bioavailability of drugs by oral administration [214,215,216], increasing the intracellular penetration and retention time, controlling the release of encapsulated drugs, and targeting delivery in specific cancer cells [217]. In the case of colon cancers, nanoparticles are especially useful, because they can help to overcome the different pH of the different parts of the digestive tract and to protect drugs from degradation, premature drug release, and absorption before reaching the colon.

Critical issues for the efficacy of these systems in treating CRC are the cellular uptake, accumulation, and clearance of circulating nanoparticles. Nanocarriers are accumulated in the tumour tissue due to the disruption of the vascular barrier [218,219]. The gap between the endothelial cells in the tumour vasculature depends on the tumour type, localisation and environment and is responsible for this drawback. Furthermore, the poor lymphatic function of the tumour tissues impedes the clearance of nanoparticles from the tumour interstitium [218,220] known as the enhanced permeability and retention (EPR) effect, which is the basis for passive targeting [218,221]. This build-up of the drug at the tumour sites is a passive process. Therefore, it necessitates extended motion of the drug for proper drug delivery. 

The accumulation of the nanocarriers is fundamentally determined by their physicochemical properties such as size, shape (morphology), surface charge and surface chemistry [218,222]. The extent and kinetics of nanomaterial accumulation at the tumour site are largely influenced by their size. Therefore, the nanocarriers need to be smaller than the cut-off of the proportions in the neovasculature, with the extravasation to the tumour acutely affected by the size of the vehicle. 

Nanoparticle shapes also influence their accumulation in cancer cells, and the best shape to encourage longer circulation time and faster uptake by cancer cells was found to be the spherical one [223,224]. Other relevant parameters include the optimal surface charge to promote cellular binding and prevent complement activation (range: 0 to −10 mV) [223,225] and the sufficient density of targeting ligands on the surface of nanoparticles to optimise tissue-specific targeting (range: 0.5–5%). Further, the biodistribution of the nanomaterial–drug formulation is influenced by blood perfusion, passive interactions with biomolecules along the route, and immunological clearance processes, such as phagocytosis or renal clearance [219,226,227,228].

Despite major advances in the use of nanoparticles as therapeutic platforms for the treatment of prostate [229], ovarian [230], breast [231,232] and lung cancers [233,234], the clinical development of nanoparticles for the treatment of CRC remains limited.

GO is a promising nanomaterial with enormous possibilities in cancer therapeutic approaches due to its unique and unequalled structure and properties. Its main applications in anticancer therapy are as a nanocarrier and photosensitizer. These GO properties are yet in the preliminary stages of their investigations, with many issues waiting to be resolved. One of them is GO’s effectiveness and safety which need further research. Among the reasons for the poor effectiveness of GO is its tendency to aggregate under physiological conditions [235,236]. The lack of enough stability in the biological medium can hinder cancer photothermal therapy using GO-based materials. Surface modifications alter GO’s physical and chemical properties and biocompatibility and, thus, can improve the drug carrier and photothermal efficiency. Covalent modifications of GO often require multi-step chemical reactions, which may affect the activity of biomolecules, unlike noncovalent modification, which is a simpler process [235,237]. Various polymers have been used for the functionalisation of GO and have been shown to improve the colloidal stability, cytotoxicity and size distribution, as well as biocompatibility and cellular uptake [238]. However, all types of functionalized GO nanostructures still need to be further examined, because each form of GO-based material possesses its exclusive properties and surface functionalities, which influence the material’s behaviour in the body [239]. 

To date, GO and its derivatives have been profoundly characterised by their In vitro biological performance. What happens inside the body after loading the GO carriers has not been adequately studied. The controlled release of the loaded drugs from the GO carriers is still immature, despite the efforts of many authors invested in dissecting the way GO and its different formulations control drug release. The current GO nanocarriers load drugs mainly by noncovalent interactions, which makes the GO–drugs complex unstable [240]. Additionally, GO could passively accumulate in cancer tissue due to the EPR effect [235]. More attention has to be paid to active targeting, which could improve the clinical application of GO. However, GO targeting still needs to be improved and optimised. The data about the elimination of GO and its derivates from the body or its biological clearance, as well as the relation between clearness and the chemical modifications of GO are very scarce [241]. GO is mainly used to load small drug molecules but not biomacromolecules such as DNA and proteins [242,243]. Further, non-targeted drug delivery by GO and its derivates leads to nonspecific distribution and the uncontrollable release of the drug, thus decreasing the effectiveness of cancer treatment. Therefore, researchers have to pay more attention to the design of innovative hybrid GO nanostructures that can deliver drugs to the target sites with a reduced dosage frequency and in a spatially controlled manner to mitigate the side effects. The last makes targeted drug transport a crucial task for increasing the GO potential in cancer mono- or combined therapies, particularly for colorectal carcinoma cancer.

One of the major concerns is the toxicity of GO nanocarriers. Although many efforts have been made to reduce the toxicity of graphene and graphene-based materials, the use of these compounds is currently associated with a high risk. Among the critical factors for the biosafety of the nanomaterials for biomedical application are shape, size, exposure type and time, agglomeration, corona effect concentration, and impurities. For example, smaller GO flakes showed, in general, less toxicity. The size of GO is also important to be always taken into account, as a mechanism by which the material internalises into a target cell [244,245,246]. 

The interaction between graphene nanomaterials and cells/tissues is the main toxicity issue. Synthetic and purification methods must be carefully examined, because, in many studies, toxicity is attributed to the contaminants present in the sample. The usage of hazardous or toxic compounds should be avoided in the process of designing GO-based anticancer systems, preventing possible adverse health effects, skin irritations, immune reactions, and toxicity. Green and sustainable synthesis methods, as well as green functionalisation processes, can be used for the preparation of GO-based nanosystems to reduce and prevent potential environmental and health risks, as well as to enhance biocompatibility and sustainability. Rigorous cleaning processes are undeniably needed during top-down strategies for producing GO-based materials. For example, during the preparation of GO derivates, if it is not properly cleaned, tissue inflammation might occur.

Conducted preclinical studies are not enough, and a great deal of preparatory study still needs to be performed to present GO-based materials as a successful therapeutic cancer approach. Therefore, clinical and long-term studies are very important for the assessment of the long-term cytotoxicity of GO-based materials and their effects on cell signalling. Importantly, the understanding of the fine mechanisms responsible for GO toxic effects is a prerequisite for providing functionalized GO-based materials with high biocompatibility. Therefore, the design and selection of rational criteria for cytotoxicity studies of GO are of immense importance for the development of clinically successful and translatable nanomedicines. Additionally, immunogenicity, inflammatory reactions and hemocompatibility are dynamic standards for the anticancer employment of GO-based materials [7]. Data show that GO-based nanocomposites could exert DNA or mitochondrial damage, inflammatory reactions, autophagy, necrosis and apoptosis [247]. Recently, the haemolytic effects of GO structures were typically initiated via electrostatic interactions between GO and the red blood cell membrane, which can be circumvented by suitable surface functionalization or modification to improve the haemocompatibility [248]. It was revealed that GO caused significant immunogenicity, as confirmed by a notable increase of TNF- α (the tumour necrosis factor), Il-6 and Il-1(interleukin-6and 1). Nonetheless, the functionalized GO structures illustrated improved immunological compatibility [249,250].

The development of new agents for the treatment of cancer, as well as the detailed elucidation of the mechanisms of action and toxicity of the newly developed drugs in various cancers and normal cells, is of great importance for addressing the societal challenges affecting the health and quality of life of humans. Undoubtedly, studies aimed at exploring in detail the effects of the targeted release of newly synthesised GO-based drugs in combination with photothermal treatment will help to clarify the mechanism of antitumour action of possible combined cancer therapy (NP combined with photodynamic therapy). They will contribute to the understanding of the fine molecular mechanisms of action of this innovative approach on cell proliferation, adhesion, ROS generation, DNA damage, and cell cycle influences of the cells under study. In addition, comparing the effects of such a combination therapy on normal and tumour cells will lead to the accumulation of new data and, accordingly, the development of strategies for innovative cancer-targeted therapies.

## Figures and Tables

**Figure 1 pharmaceutics-14-01213-f001:**
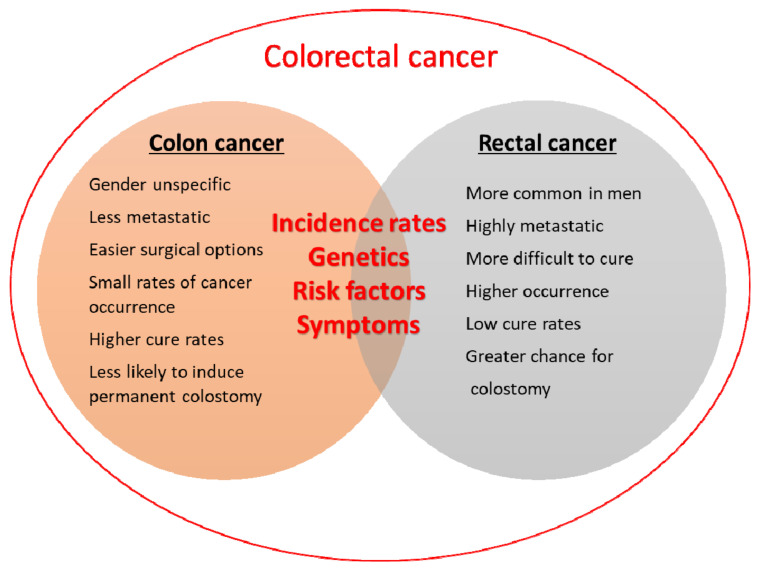
Comparison of colon and rectal cancers.

**Figure 2 pharmaceutics-14-01213-f002:**
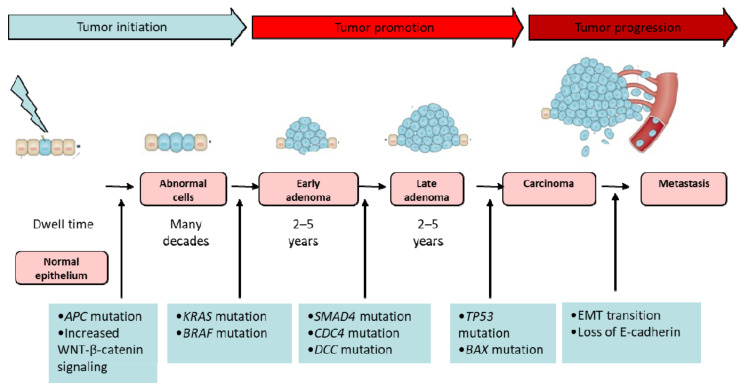
Mechanisms of colorectal tumorigenesis.

**Figure 3 pharmaceutics-14-01213-f003:**
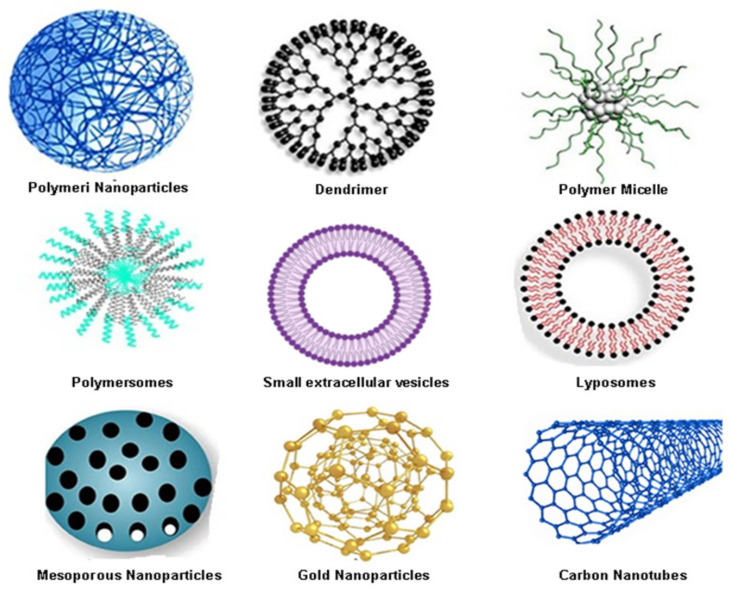
Types of nanomaterials in laboratory, preclinical, and clinical oncology practice. Images are adapted from [76].

**Figure 4 pharmaceutics-14-01213-f004:**
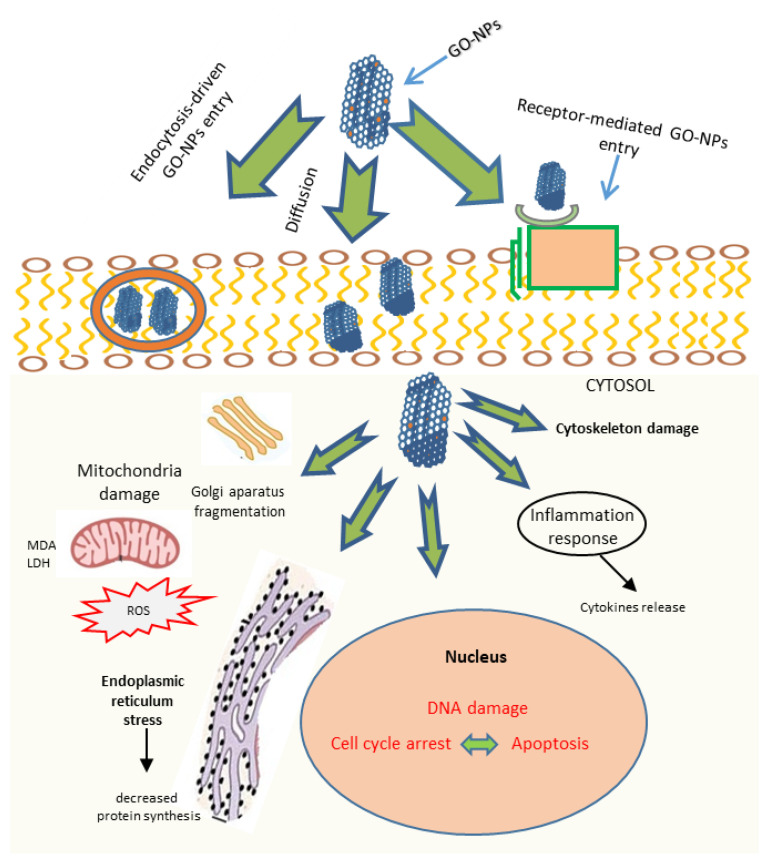
Schematic diagram of the molecular pathways for nanoparticle delivery to cancer cells and potential antitumor effects.

**Figure 5 pharmaceutics-14-01213-f005:**
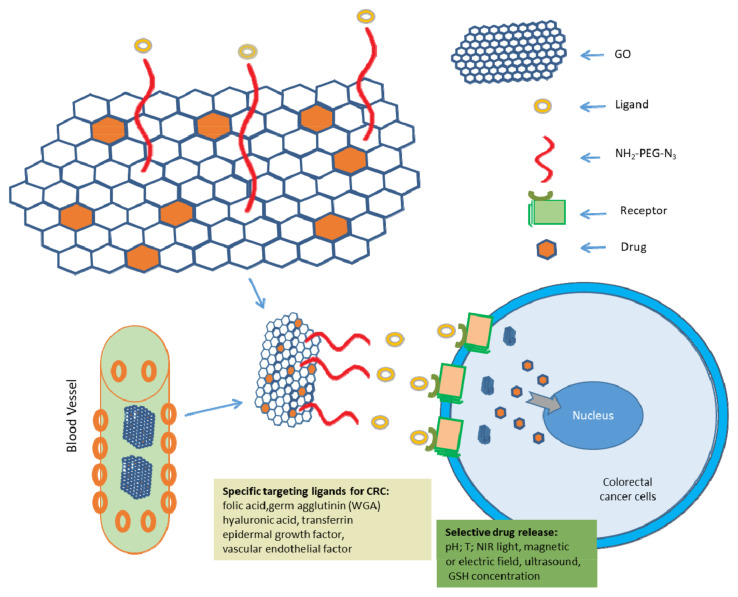
Schematic diagram of the mechanism of targeted drug delivery in colorectal cancer cells. Graphene oxide nanoparticle (GO) NPs are loaded with the drug and a specificlig and to bind specifically to colorectal cancer cells. Drugs released from GO carriers in CRC cells are triggered internally by cellular changes in the pH and redox or exogenous stimuli such as near-infrared (NIR) light, ultrasound, and magnetic and electric fields.

**Table 1 pharmaceutics-14-01213-t001:** Staging for colorectal cancer and types of corresponding therapeutic approaches.

Colorectal Cancer (CRC) Staging and Treatment Approaches		
TNM	(T) How far the tumour has grown through the colon/rectal wall (N) The presence of cancer in lymph modes (M) Whether cancer metastasised (spread) to the other parts of the body apart from the colon/rectum	
Stages/Types of treatment	**Stage 0** Local treatments: surgical resection only	The earliest stage of colorectal cancer, known also as carcinoma “in situ” or intramucosal carcinoma, meaning the cancer is in the mucosa (moist tissue lining the colon), and it is not grown through the colon/rectum wall.
	**Stage I** Local treatments: surgical resection of the malignant polyp or surgical procedure/partial colectomy of tumour and local lymph nodes	Cancer is still in the inner lining but has grown through the mucosa (second (T1) or third (T2) layer) of the colon and invaded the muscle layer. It has not spread to nearby lymph nodes (N0) or distant sites (M0).
	**Stage II** Local and systematic treatments: surgical resection without chemotherapy; chemotherapy (5-FU, leucovorin, oxaliplatin, or capecitabine)	Cancer has grown beyond the wall of the colon or rectum (T3) and is attached to or has grown into other nearby tissues or organs (T4b). It has not spread to nearby lymph nodes (N0) or distant sites (M0).
	**Stage III** Local, systematic, and combined treatments:surgery followed by adjuvant chemotherapy, including FOLFOX (5-FU, leucovorin, and oxaliplatin) or CapeOx (capecitabine and oxaliplatin) Surgery, Neoadjuvant chemotherapy, and Radiation—for some advanced colon cancers, radiation therapy and/or chemotherapy—for people who are not healthy enough for surgery	Cancer has spread from the colon/rectum to nearby lymph nodes (N2a), or there are small tumour deposits in the fat around the colon/rectum. It has not spread to distant sites (M0).
	**Stage IV** Local, systematic, and combined treatments: Radiotherapy; chemotherapy with FOLFOIRI; FOLFIRI (5-FU, LV, and irinotecan); FOLFOX; CAPIRI (capecitabine and irinotecan); CAPOX; 5-FU with LV; irinotecan; capecitabine and Trifluridine plus Tipiracil (Lonsurf); immunotherapy (Pembrolizumab (Keytruda) or Nivolumab (Opdivo); targeted therapies; palliative surgery/stenting; radiofrequency ablation; radioembolization	Cancer has metastasised to distant sites (N2a) and has been carried through the lymph and blood systems to distant parts of the body. The most likely organs to develop metastasis from colorectal cancer are the lungs and liver.

**Table 2 pharmaceutics-14-01213-t002:** Advantages and disadvantages of nanoparticles as cancer-drug delivery systems.

Nanoparticles Type	Advantages	Disadvantages
Organic nanoparticles		
Polymeric nanoparticles	Biocompatible and biodegradable; ability to entrap both hydrophilic and hydrophobic drugs; easy to modify; controlled drug release; protect the drug from metabolic degradation; prolonged residence time—bio-adhesive properties; good tissue penetration; easy manipulation; stability of drug; delivery of a higher concentration of drug to the desired location; easy merged into other activities associated to a drug delivery	Burst effect; limited drug loading capacity; high cost; low cell affinity; toxicity of degradation products; nondegradable polymers tend to accumulate in tissue; promote allergic reaction; in vivo metabolism and elimination is not elucidated; rapid clearance out of the abdominal cavity; toxic, reactive residues, unreacted monomers increase the risk of chemical reactions and the formation of unwanted oligomers
Dendrimers	Lower polydispersity index; the outer surface of dendrimer has multiple functional groups; they can be designed and synthesised for a specific application	High cost of synthesis process; non-specific toxicity; low loading capacity
Polymeric micelles	Prolonged retention time; easily synthesis; can be coupled with targeting ligands to increase accessibility to tumour sites, reduce the side effects; ability to control drug dissemination over a long period	Increased systemic toxicity
Polymersomes	Chemical versatility; an ability for controlled release and improved cellular uptake of anti-cancer molecules; low toxicity	Toxicity risk of polymers or metabolites; Polymer aggregation; hydration may be a challenge
Small extracellular vesicles	Biocompatible; safe degradation products; non-toxic; non-immunogenic; possibility for cell targeting	Low water solubility
Liposomes	Increase the efficacy and therapeutic index of drugs; biocompatible and completely biodegradable; low toxicity; flexible; improved pharmacokinetics of cargo; able to entrap both hydrophilic and hydrophobic drugs; controlled release protects the drug from metabolic degradation prolonged residence time—precorneal and vitreous; decreased the exposure of sensitive tissue to toxic drugs	Poor stability; could crystallise after prolonged storage conditions; difficult to prepare and sterilise; high cost; poor or moderate drug loading capacity; immunogenicity; low solubility; short half-time; leakage and fusion of encapsulated drug/molecules
Inorganic nanoparticles		
Mesoporous silica nanoparticles	High drug and genes loading capacity; tuneable pore size; large surface area; biocompatible and biodegradable; controlled porosity; versatility; non-toxic; easy endocytosis, and resistance to heat and pH	Expensive; not enough information about cytotoxicity, biodistribution, biocompatibility’ low stability formation of aggregates, haemolysis
Metallic and magnetic nanomaterials	Easy preparation and functionalisation; large surface area; multimodal application; high surface area; multiple forms (spherical, nanorod, triangles); biocompatibility; tuneable size; easy functionalisation excellent biodegradability in vivo; no leakage of encapsulated drugs	Low stability and storage; not enough information about uptake, biocompatibility, and low cytotoxicity *in vivo*
Carbon-based nanomaterials		
Carbon nanotubes	Water-soluble; multifunctional; less toxic; biocompatibility; biodegradability; able to entrap both hydrophilic and hydrophobic drugs; high loading capacity; a high number of possibilities for surface modification; high surface area, needle-like structure, heat conductivity, and chemical stability	Expensive to produce; low degradation; not enough *in vivo* studies

**Table 3 pharmaceutics-14-01213-t003:** Nanoparticle-based nanoformulations were used in in vitro and in vivo colorectal cancer studies.

**Type of Nanoparticles**	**Delivered Anticancer Drug/Active Compound**	**Type of CRC Cells/Tumour**	**Highlights of the Study**	**References**
PGLA NPs	Curcumin	HT-29	Increased cellular uptake than pure curcumin	[77]
PGLA NPs	SN-38	COLO-205	Improved solubility, stability, and cellular uptake; reduced cell proliferation and death	[79]
PEG-Telodendrimers	Vitamin E, Cholic acid, Gambogic acid (GA)	HT-29	A superior in vitro cytotoxic activity compared to the free drug	[82]
PEG-PES- dendrimers	PES-Polyester Doxorubicin	C-26; BALBc mice with s.c. C-26 tumours	Dendrimer–DOX was 10 times less toxic than free DOX towards C-26 cells after 72 h-exposure nine-fold higher tumour uptake than i.v. administered freeDOX at 48 h. Incomplete tumour regression and 100% survival of the mice over the 60-day experiment. Enhanced blood circulation time, reduced toxicity and less accumulation of drugs prevent the development of Dox-unresponsive C-26 tumour	[87]
PR_b-PEO-PMCL and PR_b-PEO-PMCL polymersomes	Cisplatin	DLD-1 overexpressing α5β1 integrin	Increased delivery efficacy to DLD cells; Decreased colon cancer cell viability specific targeting to α5β1 overexpressing cancer cells	[101]
HA-MSNs	Doxorubicin	HCT-116	Increased cellular uptake of DOX-HA-MSNs conjugate; enhanced cytotoxicity in cancer cells	[117]
MPVA-AP1	Doxorubicin, Vitamin B12, Curcumin	CT26-IL4R	Excellent selectivity and targeting toward CT26-IL4R cells; rapid and controlled drug release upon treatment with a high-frequency magnetic field	[122]
PLGA-coated iron NPs	5-fluorouracil (5-FU)	HT-29	Greater DNA damage in an HT-29 colon tumour cell line as compared with hyperthermia	[123]
PTX-SPIO-PEALCa micelles	Paclitaxel, Super-paramagnetic iron oxide (SPIO)	CRCLoVo	Inhibition of CRC LoVo cell growth	[124]
PDM-SWCNTs	Paclitaxel	Caco-2, HT-29	Enhanced anticancer effects in Caco-2 and HT-29 cells compared to paclitaxel alone; induced disruption of tight junctions in Caco-2 cells	[128]
MWCNTs	Oxaliplatin, Mitomycin C induced by infrared light rays	RKO, HCT 116	Increased cell membrane permeability due to hyperthermia; reduction in cancer cell viability	[130]
TRAIL -SWCNTs		carcinoma cell lines	Ten times increased cell death in comparison with alone TRAIL	[130]

**Table 4 pharmaceutics-14-01213-t004:** Grapheneoxide-based nanoformulation for application in colorectal cancer therapies.

Nanocarrier	Functionalisation Agent/Drug	Cancer Cell Line	Type of Study	Highlights of the Study	Reference
*Chemotherapy*					
CS-(NaPO4)3/rGO	Chitosan (CS), 5-FU, Curcumin	HT-29	*in vitro*	Higher entrapment efficiency (>90%); Increased cytotoxicity, IC_50_ of 23.8μg/mL for dual-drug-loaded nanocomposite; Synergistic cytotoxicity	[186]
NGO-PEG	SN38	HCT-116	*in vitro*	IC_50_ of ~6 nM ~1000-fold more potent than CPT-11 and similar to that of free SN38	[15]
GO-F38, GO-T80 and GO-MD	Ellagic acid (EA)	HT29	*in vitro*	Increased antitumour activity	[161]
AuNPs-rGO	Curcumin	HT-29, SW-948	*in vitro*	Increased ROS and cytotoxicity	[185]
GO–N=N–GO/PVA	Curcumin	mice	*in vivo*	Improved bioavailability of curcumin and preferential accumulation in the colon	[193]
*Phototherapy*					
GT-rGO sheets	Green tea	SW48, HT29	*in vitro*	20% higher photothermal destruction of the high metastatic SW48 cancer cells than that of the low metastatic HT29 cells	[179,204]
*Phototherapy/Chemotherapy*					
rGO-PDA@MS/HA	Mesoporous silica (MS), Hyaluronic acid (HA), Polydopamine (PDA), Ce6	HT-29, HCT-116	*in vitro*		[200]
GO-HA-Asp	Irinotecan, Hyaluronic acid/Polyaspartamide	HCT 116	*in vitro*	Synergistic hyperthermic/cytotoxic effect	[192]
Chit-rGO-IR-820	DOX, Chitosan, IR-820	C26	*in vitro*	Synergistic anticancer activity (chemotherapy, PTT and PDT)	[207]
*Phototherapy/Immunotherapy*					
PEG-rGO-FA-IDOi	IDO inhibitor (IDOi), Folic acid, PEG	CT26	*in vitro*, *in vivo*	Synergistically immune/PTT antitumour response	[205]
*Direct Killing/Chemosensitizing effect*					
GO		HCT116	*in vitro*	Disrupted microtubule assembly; Arrested cells in the S phase with increased accumulation in Sub-G1 population of cell cycle; induces apoptosis by generating reactive oxygen species (ROS) in a dose- and time-dependent manner	[208]
GO, GO-NH_2_		C26	*in vitro*	Induced ROS production and blocked cell cycle in the G0-G1	[210]
GO		HCT116	*in vitro*	Inhibited migration and invasion of cancer cells by inhibiting the activities of electron transport chain (ETC) complexes present in mitochondria; reduced ATP synthesis; inhibited F-actin cytoskeleton assembly	[212]
GO		CT26, mice	*in vitro*, *in vivo*	Induced Toll-like receptors (TLRs) responses and autophagy	[213]

## Data Availability

Not applicable.

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
