# Peer review of "Promising Therapeutic Strategies for Colorectal Cancer Treatment Based on Nanomaterials"

_pharmaceutics, 2022, doi:10.3390/pharmaceutics14061213_

Round 1

Reviewer 1 Report

This review summarized different types of nanomaterials for colorectal cancer. The use of graphene oxide and graphene oxide derivates in colorectal cancer was emphasized, and the outlook of this kind of material was also proposed. This review cites lots of references and has certain scientific significance. However, some of the issues mentioned below prevent me from recommending this·paper published in Pharmaceutics directly.

Recommendation: minor revision

  1. Many references are cited in this paper, and it is relatively obscure to understand only with the textual description. It would be better if the authors provide the figures about the structure of the nanomaterials as well as their anti-tumor effects.
  2. The authors should conduct an in-depth dissection of the future trends and ways to improve the effect of treatment of graphene oxide and graphene oxide derivates in the Outlook section.
  3. Note that spaces and paragraph indents should be consistent in the whole paper. For example, there is no first line indent in “1. Overview of the current situation worldwide” and “2. Colorectal cancer – an old man's disease” has the first line indent.
  4. Pay attention to the use of capitalization and italics throughout the paper. For example, on line 288, “In Common treatment” should be “In common treatment”.
  5. Note the order in which the references are listed. Number 45 is repeated, and all the number appears with two different forms.

Author Response

Herein, we express our gratitude for all comments made by the reviewer. We have followed all of them adequately and have edited Review version 1 of our manuscript. All changes are marked with track changes.

 Comments and Suggestions for Authors

This review summarized different types of nanomaterials for colorectal cancer. The use of graphene oxide and graphene oxide derivates in colorectal cancer were emphasized, and the outlook of this kind of material was also proposed. This review cites lots of references and has certain scientific significance. However, some of the issues mentioned below prevent me from recommending this paper published in Pharmaceutics directly.

Recommendation: minor revision

Many references are cited in this paper, and it is relatively obscure to understand only with the textual description. It would be better if the authors provide the figures about the structure of the nanomaterials as well as their anti-tumour effects.

We have provided additional figures that illustrate the material we discuss in the text. Please, see the revised version of our work.

We understand the concern of the reviewer for the numerous references, but the topic is a sea of NPs applications and we feel obliged at least to provide some of them here.

The authors should conduct an in-depth dissection of the future trends and ways to improve the effectiveness of treatment of graphene oxide and graphene oxide derivates in the Outlook section.

This topic has been discussed in many places in the paper. Following this recommendation, we have highlighted this information in the Conclusions and perspectives.

Note that spaces and paragraph indents should be consistent in the whole paper. For example, there is no first line indent in “1. Overview of the current situation worldwide” and “2. Colorectal cancer – an old man's disease” has the first line indent.

We have edited and proofread our paper and have spell checked it too. We now believe that all these mistakes are omitted.

 Pay attention to the use of capitalization and italics throughout the paper. For example, on line 288, “In Common treatment” should be “In common treatment”.

We have corrected this in the new version of the paper. Thank you very much for this remark.

Note the order in which the references are listed. Number 45 is repeated, and all the number appears in two different forms.

Already this repetition is solved.

Reviewer 2 Report

In this article, the authors reviewed the therapeutic strategies for colorectal cancer treatment based on nanomaterials. Authors tried to provide a summary of current advances in nanoparticle-based targeted therapy against CRC. Given the significant morbidity and mortality of CRC and promising advances in this field of nanoparticle-based targeted therapy this review could be an interesting read for researchers active and passionate about finding a way to fight this dreadful disease. I’ll recommend this manuscript for publication in "Pharmaceutics" journal after minor revision. Listed here my recommendations:

  • Abstract: abstract final paragraph misleads readers and doesn’t match the title and what the major part of review made of in my opinion.

“This review presents an update on recent advances in cancer therapies using graphene oxide and graphene oxide derivates and pays special attention to promising alternative strategies against colorectal cancer on the bases of graphene oxide nanomaterials.”

In my opinion this review is an update on recent advances in nanomaterial based CRC targeted therapy with special attention to graphene oxide nano material.

  • Table 1- it would be very informative to readers if authors add current treatment methods for each stage in Table1
  • It is disappointing to see authors failed to cite many works published in the graphene oxide-based nanomaterials for cancer targeted therapy. This need to be addressed in revised version. Few papers listed here
  1. Bioactive Materials Volume 14, August 2022, Pages 335-349
  2. Cui et al. J Nanobiotechnol (2021) 19:211
  3. Varma, et al DOI: 10.1002/mco2.118
  • Authors divided the nanostructured materials to four types. I suggest authors clearly state which type of nanostructure graphene oxide belong to for the readers that not familiar with graphene
  • There are few unmatching font size that needs to be corrected  

Author Response

Dear Reviewer,

Thank you indeed for all your comments and remarks. They, indeed, allowed us to increase the quality of our paper.

All comments are taken into consideration and the paper is edited accordingly. Please, see the revised version with all changes highlighted by track changes.

Comments and Suggestions for Authors

In this article, the authors reviewed the therapeutic strategies for colorectal cancer treatment based on nanomaterials. Authors tried to provide a summary of current advances in nanoparticle-based targeted therapy against CRC. Given the significant morbidity and mortality of CRC and promising advances in this field of nanoparticle-based targeted therapy this review could be an interesting read for researchers active and passionate about finding a way to fight this dreadful disease.

I’ll recommend this manuscript for publication in "Pharmaceutics" journal after minor revision. Listed here my recommendations:

Abstract: abstract final paragraph misleads readers and doesn’t match the title and what the major part of review made of in my opinion.

“This review presents an update on recent advances in cancer therapies using graphene oxide and graphene oxide derivates and pays special attention to promising alternative strategies against colorectal cancer on the bases of graphene oxide nanomaterials.”

In my opinion this review is an update on recent advances in nanomaterial based CRC targeted therapy with special attention to graphene oxide nano material.

We agree with the reviewer and have changed the Abstract accordingly.

Table 1- it would be very informative to readers if authors add current treatment methods for each stage in Table1

We have provided current therapeutic approaches in Table 1 in relation to CRC stages. Please, see the revised version.

It is disappointing to see authors failed to cite many works published in the graphene oxide-based nanomaterials for cancer targeted therapy. This need to be addressed in revised version. Few papers listed here

Bioactive Materials Volume 14, August 2022, Pages 335-349

Cui et al. J Nanobiotechnol (2021) 19:211

Varma, et al DOI: 10.1002/mco2.118

We agree and apologize for this. These references are already included in the text.

The authors divided the nanostructured materials to four types. I suggest authors clearly state which type of nanostructure graphene oxide belong to for the readers that not familiar with graphene

 We have provided an introductory paragraph before the part dealing with GO and there we state clearly the belonging of GO to the carbon NPs. Thank you for this valuable suggestion.

There are few unmatching font size that needs to be corrected  

Already edited in the text.

Reviewer 3 Report

Comments: Manuscript is interesting, well written and it is strongly recommended to be published in pharmaceutics after addressing of these few issues.

  • Please check the manuscript thoroughly for English typos Errors.
  • Colon cancer and colorectal cancer are so difference. Authors should to clarify this difference and write deep description from side of histology, physiology, and anatomy.
  • Authors should to provide scheme illustrates mechanism of targeted therapies to colorectal cancer.
  • The authors should discuss advantages and disadvantages of nanomaterials, as well as potential side effects. Discuss why nanomedicines are advantageous in colorectal drug delivery, and how their accumulation may depend upon the properties of the nanoparticles (e.g. size, surface charge, shape)
  • Authors have to highlight the putative benefits of targeted nano-therapies using ligands or specific molecules that can target colorectal cancer cells
  • The manuscript is missing a comprehensive discussion If some formulations have reached clinical application, the authors should state it clearly and discuss the clinical results achieved.
  • Authors should to provide the recent work of colorectal animal model treated by GO and its derivatives

Author Response

Dear Reviewer,

Thank you very much for these comments and remarks. We have addressed all of them in the newly edited version of our paper. All changes are marked with track changes.

Comments: Manuscript is interesting, well written and it is strongly recommended to be published in pharmaceutics after addressing of these few issues.

Please check the manuscript thoroughly for English typos Errors.

Already full and grammarly spell check and proofread is performed.

Colon cancer and colorectal cancer are so difference. Authors should to clarify this difference and write deep description from side of histology, physiology, and anatomy.

We have added a figure and an explanatory text that deal with the similarities and dissimilarities of these two types of cancers (Figure 1).

Authors should to provide scheme illustrates mechanism of targeted therapies to colorectal cancer.

We have provided a figure that deals with the NPs-based targeted mechanisms of CRC therapeutic approaches. Please see Figure 5. Thank you for this remark!

The authors should discuss advantages and disadvantages of nanomaterials, as well as potential side effects. Discuss why nanomedicines are advantageous in colorectal drug delivery, and how their accumulation may depend upon the properties of the nanoparticles (e.g. size, surface charge, shape)

We have done this and have provided an additional table with these data too. Please see Table 2.

The accumulation of NPs is discussed in the Conclusions section.

Authors have to highlight the putative benefits of targeted nano-therapies using ligands or specific molecules that can target colorectal cancer cells

We have added these data in Section: 2.2.

The manuscript is missing a comprehensive discussion If some formulations have reached clinical application, the authors should state it clearly and discuss the clinical results achieved.

These data are already added in the new version of the review. Please, see 1.6. and lines 338, etc.

Authors should to provide the recent work of colorectal animal model treated by GO and its derivatives

Table 3 summarizes these data.

Round 2

Reviewer 3 Report

Authors have revised the comment of reviewer point by point and manuscript is more acceptable  now.